# A reference haplotype panel for genome-wide imputation of short tandem repeats

Shubham Saini [1], Ileena Mitra [2], Nima Mousavi [3], Stephanie Feupe Fotsing[2,4] & Melissa Gymrek [1,5]

Short tandem repeats (STRs) are involved in dozens of Mendelian disorders and have been implicated in complex traits. However, genotyping arrays used in genome-wide association studies focus on single nucleotide polymorphisms (SNPs) and do not readily allow identification of STR associations. We leverage next-generation sequencing (NGS) from 479 families to create a SNP + STR reference haplotype panel. Our panel enables imputing STR genotypes into SNP array data when NGS is not available for directly genotyping STRs. Imputed genotypes achieve mean concordance of 97% with observed genotypes in an external dataset compared to 71% expected under a naive model. Performance varies widely across STRs, with near perfect concordance at bi-allelic STRs vs. 70% at highly polymorphic repeats. Imputation increases power over individual SNPs to detect STR associations with gene expression. Imputing STRs into existing SNP datasets will enable the first large-scale STR association studies across a range of complex traits.

[1] Department of Computer Science and Engineering, University of California San Diego, 9500 Gilman Drive, La Jolla, CA 92093, USA. [2] Bioinformatics and Systems Biology Program, University of California San Diego, 9500 Gilman Drive, La Jolla, CA 92093, USA. [3] Department of Electrical and Computer Engineering, University of California, San Diego, 9500 Gilman Drive, La Jolla, CA 92093, USA. [4] Department of Biomedical Informatics, University of California San Diego, 9500 Gilman Drive, La Jolla, CA 92093, USA. [5] Department of Medicine, University of California, San Diego, 9500 Gilman Drive, La Jolla, CA 92093, USA. Correspondence and requests for materials should be addressed to M.G. (email: mgymrek@ucsd.edu)

Genome-wide association studies (GWAS) have become increasingly successful at identifying genetic loci significantly associated with complex traits in humans, largely due to the enormous growth in available sample sizes[1–3]. Hundreds of thousands of individuals have been genotyped using commodity genotyping arrays. These arrays take advantage of the correlation structure between nearby variants induced by linkage disequilibrium (LD), which allows genome-wide imputation based on genotypes of only a small subset of loci[4]. However, GWAS based on single-nucleotide polymorphism (SNP) associations face important limitations. Even with sample sizes of up to 100,000 individuals, common SNPs still fail to explain the majority of heritability for many complex traits[1,5].

One compelling hypothesis explaining the missing heritability dilemma is that complex variants, such as multi-allelic repeats not in strong LD with common SNPs, are important drivers of complex traits but are largely invisible to current analyses. Indeed, dissection of the strongest schizophrenia association, located in the major histocompatibility complex, revealed a poorly tagged polymorphic copy number variant (CNV) to be the causal variant[6]. The signal could not be localized to a single SNP and could only be explained after deep characterization of the underlying CNV. This and subsequent discoveries[7,8] highlight the importance of considering alternative variant classes.

Short tandem repeats (STRs), consisting of repeated motifs of 1–6 bp in tandem, comprise more than 3% of the human genome[9]. Multiple lines of evidence support a role of STRs in complex traits[10–12], particularly in neurological and psychiatric phenotypes. Due to their rapid mutation rates[13], STRs exhibit high rates of heterozygosity[14] and likely contribute at least as many de novo mutations per generation as SNPs[15,16]. Furthermore, STRs have been shown to play a significant role in regulating gene expression[17,18], splicing[19–21], and DNA methylation[18]. Intriguingly, more than 30 Mendelian disorders are caused by STR expansions via a range of mechanisms, including polyglutamine aggregation (Huntington's Disease, ataxias[22]), hypermethylation (Fragile X Syndrome[23]), and RNA toxicity (ALS/FTD[24]). Furthermore, causal STRs driving existing GWAS signals have already been identified[25].

Existing technologies have not allowed for systematic STR association studies. Next-generation sequencing (NGS) can be used to directly genotype short STRs, but NGS is still too expensive to perform on sufficiently large cohorts for GWAS of most complex traits. An alternative approach is to impute STRs into existing SNP array datasets. Previous studies have demonstrated that STRs are often in significant LD with nearby SNPs[26–28] and found that STRs and SNPs provide complementary information about the evolutionary history of a genomic region. Despite widespread SNP-STR LD, statistical phasing of STRs and SNPs is challenging for several reasons: SNP-STR LD is notably weaker than SNP-SNP LD[28] due to the rapid mutation rates[13,29] and high prevalence of recurrent mutations in STRs. As a result, the relationship between STR repeat number and SNP haplotype can be complex: the same STR allele may be present on multiple SNP haplotypes. On the other hand, a single SNP haplotype may harbor multiple distinct STR alleles. Furthermore, LD patterns at STRs vary widely as a function of properties of the repeat, such as the repeat unit length, mutation rate, and mutation step size[28]. Finally, STRs are prone to genotyping errors induced during PCR amplification[30,31], further ambiguating phase information.

Sequencing related samples allows haplotype resolution by directly tracing inheritance patterns. The recent generation of deep NGS using PCR-free protocols for hundreds of nuclear families in combination with accurate tools for genotyping STRs from NGS[32] now enables applying this technique genome-wide. Here, we profile STRs in 479 families and use pedigree information to phase STR genotypes onto SNP haplotypes to create a genome-wide reference for imputation. We use this panel to impute STRs into an external dataset of similar ethnic background with average 97% concordance with observed STR genotypes. Imputation accuracy varies across STRs, ranging from nearly perfect concordance at bi-allelic STRs to around 70% for highly polymorphic forensic markers. We show that STR imputation achieves greater power than individual SNPs to detect underlying STR associations and demonstrate the utility of our panel by detecting STRs not previously known to be associated with gene expression. Finally, we impute genotypes at STRs previously implicated in human disorders and show that we could accurately identify specific SNP haplotypes associated with long normal alleles most at risk for expansion.

To facilitate use by the community, we release a phased SNP + STR haplotype panel for samples genotyped as part of the 1000 Genomes Project (see Data availability). This resource will enable large-scale studies of STR associations in hundreds of thousands of available SNP datasets, and will likely yield significant new insights into complex traits.

## Results

**A catalog of STR variation in 479 families.** We first generated a genome-wide catalog of STR variation in a cohort of families included in the Simons Simplex Collection (SSC) (see URLs). We focused on 1916 individuals from 479 family quads (parents and two children) that were sequenced to an average depth of 30x using Illumina's PCR-free protocol. Based on comparison to 1000 Genomes Project samples, we estimated the cohort to consist primarily of Europeans (83%), with 2.0%, 9.0%, and 3.6% of East Asian, South Asian, and African ancestry, respectively (Supplementary Fig. 1). We used HipSTR[32] to profile autosomal STRs in each sample. HipSTR takes aligned reads and a reference set of STRs as input and outputs maximum likelihood diploid genotypes for each STR in the genome. While HipSTR infers the entire sequence of each STR allele, we focus here on differences in repeat copy number rather than sequence variation within the repeat itself. To maximize the quality of genotype calls, individuals were genotyped jointly with HipSTR's multi-sample calling mode using phased SNP genotypes and aligned reads as input (Methods). Multi-sample calling allows HipSTR to leverage information on haplotypes discovered across all samples in the dataset to estimate per-locus error parameters and output genotype likelihoods for each possible diploid genotype. Notably, our HipSTR catalog excluded most known STRs implicated in expansion disorders such as Huntington's Disease and hereditary ataxias, since even the normal allele range for these STRs is above or near the length of Illumina reads[33–36]. To supplement our panel, we applied a second STR genotyper, Tredparse[37], to genotype a targeted set of known pathogenic STRs in our cohort (Supplementary Table 1). Tredparse incorporates multiple features of paired-end reads to estimate the size of repeats longer than the read length. For seven STRs called by both Tredparse and HipSTR, Tredparse genotypes were used for downstream analyses.

An average of 1.14 million STRs passed HipSTR's default filtering settings in each sample (Fig. 1a). We obtained at least one call for 97% of all STRs in the HipSTR reference of 1.6 million STRs and for 15 of 25 STRs in the Tredparse reference with an average overall call rate of 90% (Fig. 1b). We applied additional stringent genotype quality filters to ensure accurate calls for downstream phasing and imputation analysis. STRs overlapping segmental duplications, with call rates <80%, or with genotype frequencies unexpected under Hardy-Weinberg Equilibrium were removed (Methods). We further removed STRs with low expected

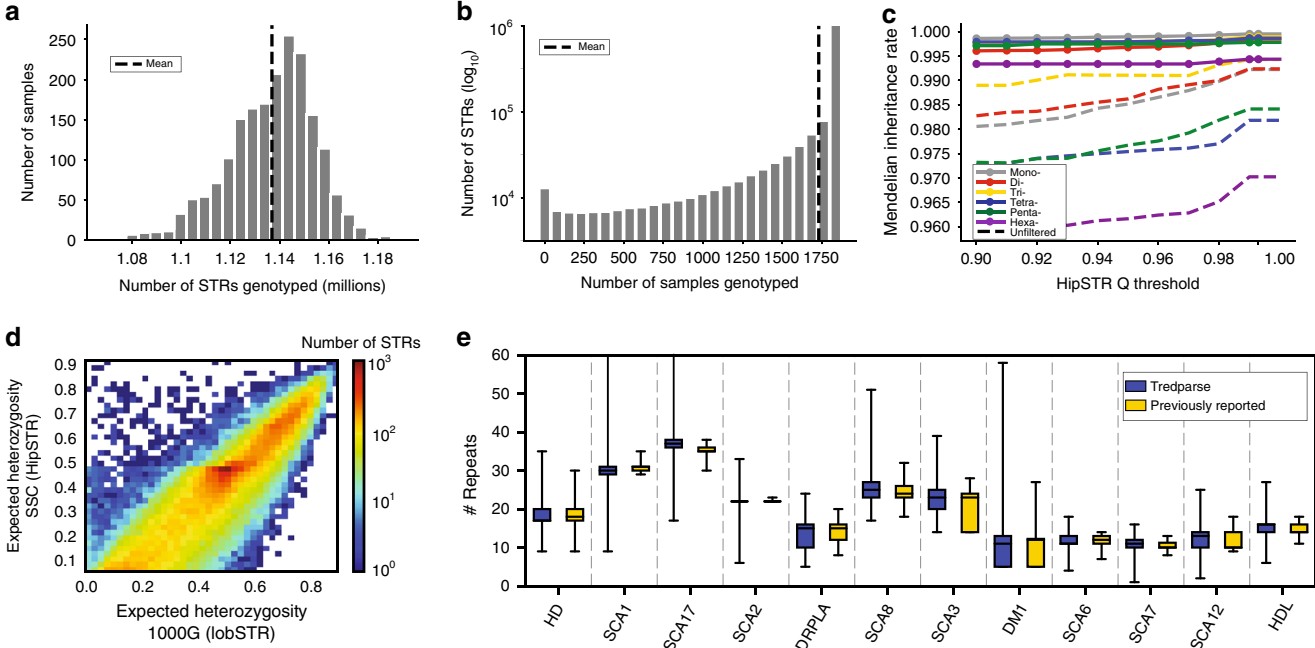

**Fig. 1** A deep catalog of STR variation in the SSC cohort. **a** Number of STRs called per sample. Dashed line represents the mean of 1.14 million STRs per sample. **b** Call rate per locus. Dashed line represents the mean call rate of 90%. **c** Mendelian inheritance rate at filtered vs. unfiltered STRs. The x-axis gives the posterior genotype score (Q) returned by HipSTR. The y-axis gives the average Mendelian inheritance rate for each bin across all calls on chromosome 21. STRs that were homozygous for the reference allele in all members of a family were removed. Colors represent different motif lengths. **d** Per-STR expected heterozygosity in SSC vs. 1000 Genomes. Only STRs with expected heterozygosity >0.095 in SSC are included. Color scale gives the $\log_{10}$ number of STRs represented in each bin. **e** Allele frequency distributions at pathogenic STRs obtained in SSC samples vs. previously reported normal alleles. Blue = SSC, Gold = Previously reported. Boxes span the interquartile range and horizontal lines give the medians. Whiskers extend to the minimum and maximum data points. The y-axis gives the number of repeat units. Sources of previously reported allele frequencies are described in detail in Methods. HD Huntington's disease, SCA spinocerebellar ataxia, DRPLA Dentatorubral-pallidoluysian atrophy, DM1 myotonic dystrophy type 1, HDL Huntington's disease-like 2

heterozygosity (<0.095) to restrict analysis to polymorphic STRs. We found that these filters increased the quality of our calls, as evidenced by the average Mendelian inheritance rate of 99.8% and 97.9% at STRs that passed and failed quality filters, respectively (Fig. 1c). After filtering, 453,671 and 9 STRs from the HipSTR and Tredparse panels, respectively, remained in our catalog.

We further assessed the quality of our STR genotypes by comparing patterns of variation from SSC to previous catalogs of STR variation obtained using a distinct set of samples and STR genotyping methods. We found that per-locus heterozygosities (Methods) were highly concordant with a catalog generated from the 1000 Genomes Project[38] data using lobSTR[39]. (Pearson $r = 0.96$; $p < 10^{-200}$; $n = 386,100$) (Fig. 1d). Allele length distributions at known pathogenic STRs observed in SSC matched closely to previously reported normal allele frequencies at each STR (Fig. 1e). For STRs genotyped both by HipSTR and Tredparse, estimated repeat lengths were highly concordant (average concordance 99.4%, Supplementary Table 1). Overall, these results show that our catalog consists of robust STR genotypes suitable for downstream phasing and imputation analysis.

**A genome-wide SNP + STR haplotype reference panel**. We examined the extent of linkage disequilibrium between STRs and nearby SNPs using two metrics. The first, termed length $r^2$, is defined as the squared Pearson correlation between STR allele length and the SNP genotype. The second, termed allelic $r^2$, treats each STR allele as a separate bi-allelic locus and is computed similar to traditional SNP-SNP LD (Methods). Similar to previous studies[28], SNP-STR LD was dramatically weaker than SNP-

SNP LD by both metrics (Supplementary Fig. 2a) with length $r^2$ generally stronger than allelic $r^2$. We additionally determined the best tag SNP (Methods) for each STR, which was on average 5.5 kb away (Supplementary Fig. 2b). Nearly all STRs were in significant LD (length $r^2$ $p < 0.05$) with the best tag SNP, suggesting that phasing would result in informative haplotypes.

We developed a pipeline to phase STRs onto SNP haplotypes leveraging the quad family structure (Fig. 2a). Based on our LD analysis, we used a window size of ± 50 kb to phase each STR separately using Beagle[40], which was recently demonstrated to perform well in phasing multi-allelic STRs[41] and can incorporate pedigree information. Resulting phased haplotypes from the parent samples were merged into a single genome-wide reference panel for downstream imputation.

We first evaluated the utility of our phased panel for imputation using a leave-one-out analysis in the SSC samples. For each sample, we constructed a modified reference panel with that sample's haplotypes removed and then performed genome-wide imputation. We measured concordance, length $r^2$, and allelic $r^2$ between imputed vs. observed genotypes at each STR, where observed refers to genotypes obtained by HipSTR or Tredparse. We additionally evaluated imputation performance under two null models where genotypes were either imputed randomly (random model) or always imputed as the most frequent diploid genotype (naive model) (Methods). Imputed genotypes showed an average of 96.7% concordance with observed genotypes, compared to 61.0% or 71.7% expected under the random and naive models, respectively (Table 1). As expected, concordance was strongest at the least polymorphic STRs (Fig. 2b, Supplementary Fig. 3a) and allelic $r^2$ was highest for the most common

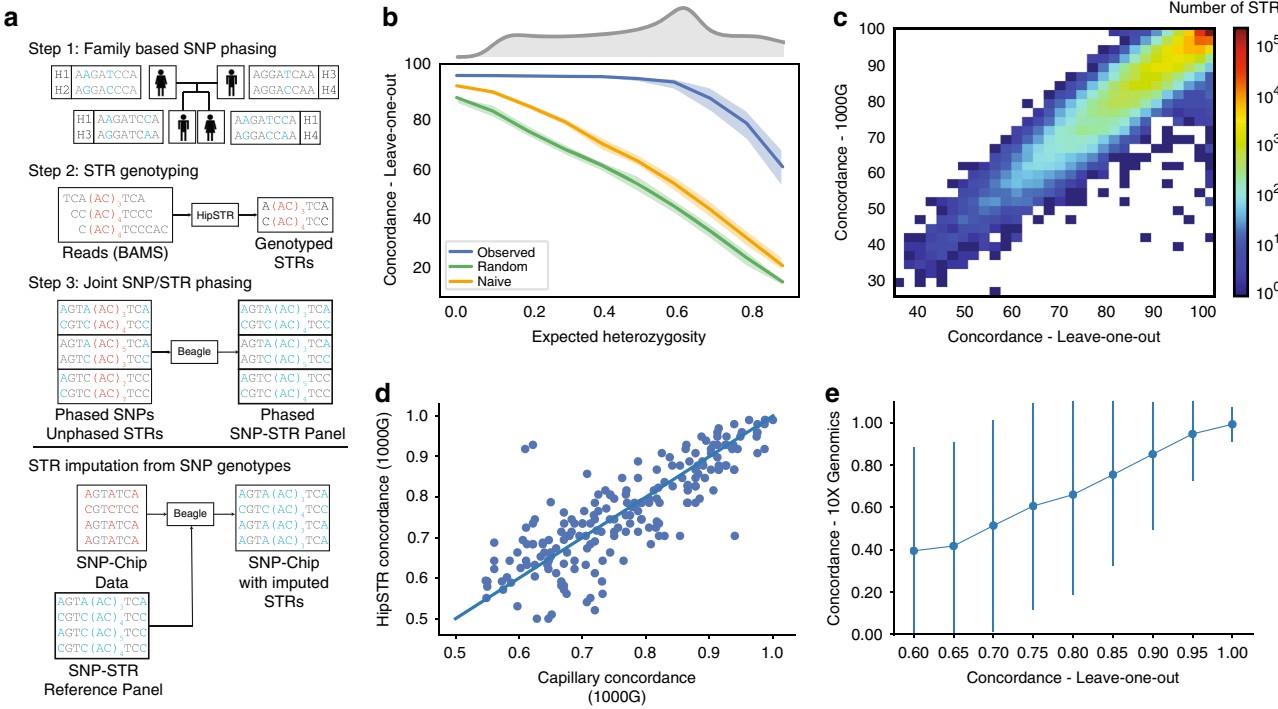

**Fig. 2** Creating a reference SNP-STR haplotype panel. **a** Schematic of phasing pipeline in the SSC cohort. To create the phased panel, STR genotypes were placed onto phased SNP haplotypes using Beagle. Any missing STR genotypes were imputed. The resulting panel was then used for downstream imputation from orthogonal SNP genotypes. Blue and red denote phased and unphased variants, respectively. Positions in gray are homozygous. **b** Concordance of imputed STR genotypes vs. expected heterozygosity. Blue denotes observed per-locus values, green denotes values expected under a random model and orange denotes values expected under a naive model. Solid lines give median values for each bin and filled areas span the 25th to 75th percentile of values in each bin. x-axis values were binned by 0.1. Upper gray plot gives the distribution of expected heterozygosity values in our panel. Concordance values are based on the leave-one-out analysis in the SSC cohort. **c** Per-locus imputation concordance in SSC vs. 1000 Genomes cohorts. Color scale gives the log₁₀ number of STRs represented in each bin. Concordance values are based on the subset of samples from the 1000 Genomes deep WGS cohort with European ancestry. **d** Per-locus imputation concordance using HipSTR vs. capillary electrophoresis genotypes. Each dot represents one STR. The x-axis and y-axis give imputation concordance using capillary electrophoresis or HipSTR genotypes as a ground truth, respectively. Concordance was measured in separate sets of 1000 Genomes European samples for each technology. **e** Concordance of imputed vs. 10X STR genotypes in NA12878 stratified by concordance in SSC. STRs were binned by concordance value based on the leave-one-out analysis. Concordance in NA12878 was measured across all STRs in each bin. Dots give mean values for each bin and lines denote ±1 s.d. In all cases leave-one-out refers to analyses performed in the SSC cohort

alleles (Supplementary Fig. 3b). Length $r^2$ was not strongly associated with expected heterozygosity, although the least and most heterozygous STRs tended to have lower length $r^2$ (Supplementary Fig. 3c). Imputation metrics were weakly negatively correlated with distance to the best tag SNP (Pearson $r = -0.06$; $p = 0.06$, Pearson $r = -0.04$; $p = 0.27$; and Pearson $r = -0.06$, $p = 7.5 \times 10^{-5}$ between distance to the best tag SNP and concordance, length $r^2$, and allelic $r^2$, respectively). To further evaluate imputation performance at highly polymorphic STRs, we examined the CODIS STRs used in forensic analysis (Supplementary Table 2). Per-STR concordances were highly correlated with imputation results recently reported by Edge et al.[41] (Pearson $r^2 = 0.93$; $p = 6.3 \times 10^{-6}$; $n = 10$), but were on average 8.8% higher (average concordance 69.1% vs. 60.3% using our panel vs. in Edge et al.[41] restricting to STRs imputed in both studies), likely as a result of our larger and more homogenous cohort. Per-locus imputation statistics for all STRs are reported in Supplementary Data 1 and 2).

We next evaluated our ability to impute STR genotypes into external datasets. For this, we focused on samples from the 1000 Genomes Project[38] with high quality SNP genotypes obtained from low coverage whole-genome sequencing (WGS) ($n = 2504$) or genotyping arrays ($n = 2486$ for Affy 6.0, and $n = 2318$ for Omni 2.5). We validated imputed genotypes for subsets of 1000

Genomes samples using data obtained from three pipelines: (1) Illumina WGS + HipSTR, (2) capillary electrophoresis, and (3) 10X Genomics + HipSTR, in each case using the orthogonal data as the truth set. Each of these datasets evaluates a different aspect of our imputation pipeline. The first tests whether a pipeline identical to that used to create our reference panel can achieve similar performance on datasets collected by different groups using different protocols. Additionally, since it consists of both Europeans and non-Europeans, it allows us to evaluate imputation across a variety of population groups. The second tests whether our results are robust across STR genotyping technologies and allows us to compare imputed STRs based on statistically inferred HipSTR genotypes to those obtained experimentally using capillary electrophoresis. The third returns phased genotypes, allowing us to directly compare inferred haplotypes and phase information.

First, we used HipSTR to genotype STRs in separate high-coverage (30×) WGS datasets available for 150 of the samples (see URLs) from European ($n = 50$), African ($n = 50$), and East Asian ($n = 50$) backgrounds. Per-locus concordance, length $r^2$, and allelic $r^2$ were highly concordant between the SSC panel and 1000 Genomes samples of European origin (Pearson $r = 0.94$, 0.63, and 0.85, respectively) (Fig. 2c; Supplementary Fig. 5; Table 1). Overall imputation performance did not vary when using phased

**Table 1 Imputation performance summary**

| Panel (n = number of samples) | Observed concordance | Naive concordance | Random concordance | Observed length $r^2$ | Random length $r^2$ | Observed allelic $r^2$ | Random allelic $r^2$ |
|---|---|---|---|---|---|---|---|
| SSC—LOO (n = 1916) | 96.7% | 71.7% | 61.0% | 0.906 | 0.605 | 0.861 | 0.552 |
| SSC—LOO (multi-allelic) | 94.3% | 62.2% | 48.5% | 0.888 | 0.334 | 0.800 | 0.333 |
| 1000 Genomes—EUR (n = 49) | 97.0% | 75.1% | 63.2% | 0.921 | 0.678 | 0.892 | 0.543 |
| 1000 Genomes—EUR (multi-allelic) | 94.8% | 66.6% | 50.0% | 0.900 | 0.334 | 0.828 | 0.314 |
| 1000 Genomes—AFR (n = 46) | 90.6% | 70.2% | 57.9% | 0.746 | 0.619 | 0.706 | 0.493 |
| 1000 Genomes—AFR (multi-allelic) | 85.6% | 61.1% | 44.4% | 0.708 | 0.336 | 0.653 | 0.310 |
| 1000 Genomes—EAS (n = 45) | 93.8% | 77.2% | 66.0% | 0.823 | 0.690 | 0.781 | 0.557 |
| 1000 Genomes—EAS (multi-allelic) | 89.4% | 69.7% | 53.7% | 0.780 | 0.336 | 0.663 | 0.313 |

Results indicate mean across all STRs analyzed. Allelic $r^2$ values include all common alleles (frequency at least 5%). Multi-allelic refers to STRs with three or more common alleles. Naive and random denote the two null imputation models as defined in the Methods

genotypes obtained from WGS vs. Omni2.5 for imputation (Supplementary Table 3). Concordance was noticeably weaker in African and East Asian samples, likely due to different population background compared to the SSC samples and lower LD in African populations[42].

Next, we compared imputed genotypes to capillary electrophoresis data[43] (see URLs) available for a subset of samples in our panel at highly polymorphic STRs. After filtering non-European samples and STRs that could not be reliably mapped to HipSTR notation (Methods), 41 samples and 206 STRs remained for comparison. We obtained an average overall concordance of 76.9% with capillary genotypes compared with 76.4% expected based on HipSTR analysis. Per-locus concordances based on HipSTR vs. capillary genotypes were strongly correlated (Pearson $r = 0.83$; $p = 1.05 \times 10^{-53}$; $n = 206$) (Fig. 2d).

Finally, we compared imputed genotypes from the highly characterized NA12878 genome to phased data available from 10X Genomics (see URLs), a synthetic long read technology. We constructed a phased validation panel by calling HipSTR separately on reads from each phase and combining with phased SNP genotypes (Methods, Supplementary Fig. 6). We could obtain phased 10X calls for 116,764 of the STRs in our panel. We used the nearest heterozygous SNP to each STR to match phase order between our panel and the 10X data, which allowed us to directly compare imputed alleles and evaluate phase accuracy. Overall, imputed STR alleles showed 96% concordance with those obtained from 10X and per-locus genotype concordance was consistent with concordance metrics measured in SSC (Fig. 2e). Taken together, validation of imputed STR genotypes against three separate truth sets demonstrates the accuracy of our original SNP + STR haplotype panel and shows that our quality metrics are reliable indicators of per-STR imputation performance across datasets.

**Imputation increases power to detect STR associations.** We sought to determine whether our SNP + STR haplotype panel could increase power to detect underlying STR associations over standard GWAS. First, we simulated phenotypes based on a single causal STR and examined the power of the imputed STR genotypes vs. nearby SNPs to detect associations. We focused primarily on a linear additive model relating STR dosage, defined as the average allele length, to quantitative phenotypes (Fig. 3a), since the majority of known functional STRs follow similar models (e.g., refs. [17,21,44,45]). Association testing simulations were

performed 100 times for each STR on chromosome 21 in our dataset (Methods). As expected, the strength of association for each variant as measured by the negative $\log_{10}$ $p$-value was linearly related with its length $r^2$ with the causal variant (Fig. 3b). On average, imputed STR genotypes explained 17.7% more variation in STR allele length compared to the best tag SNP (mean length $r^2 = 0.92$ and 0.74 for imputed STRs vs. SNPs, respectively). The advantage from STR imputation grew as a function of the number of common STR alleles (Supplementary Fig. 7). Imputed genotypes showed a corresponding increase in power to detect associations at a given $p$-value threshold (Fig. 3c). Similar trends were observed for case–control traits (Supplementary Fig. 8). We additionally tested the ability of imputed STR genotypes to identify associations due to non-linear models relating STR genotype to phenotype (Supplementary Fig. 9). While both STR and SNP-based tests had limited power to detect non-linear associations, per-allele STR association tests had higher power than the best tag SNP in 60% of simulations. Importantly, testing for complex models relating repeat length to phenotype will only be possible when allele lengths are available, thus demonstrating an additional need for STR imputation over SNP-based tests to detect these associations.

We next determined whether STR imputation could identify STR associations using real phenotypes. We focused on gene expression, given the large number of reported associations between STR length and expression of nearby genes in cis[17,18] (termed eSTRs). To this end, we analyzed eSTRs from samples in the Genotype-Tissue Expression[46] (GTEx) dataset for which RNA-sequencing, WGS, and SNP array data were available. As a test case, we imputed STR genotypes using SNP data for chromosome 21 and tested for association with genes expressed in whole blood. For comparison, we additionally performed each association using genotypes obtained from WGS using HipSTR (Methods). A total of 2452 STR x gene tests were performed in each case. Association $p$-values were similarly distributed across both analyses and showed a strong departure from the uniform distribution expected under a null hypothesis of no eSTR associations (Fig. 3d). For all nominally significant associations ($p < 0.05$), effect sizes were strongly correlated when using imputed vs. HipSTR genotypes (Pearson $r = 0.99$; $p = 1.01 \times 10^{-79}$, $n = 97$). Furthermore, effect sizes obtained from imputed data were concordant with previously reported effect sizes in a separate cohort using a different cell type (lymphoblastoid cell lines)[17] (Pearson $r = 0.79$; $p = 0.0042$, $n = 11$) (Fig. 3e).

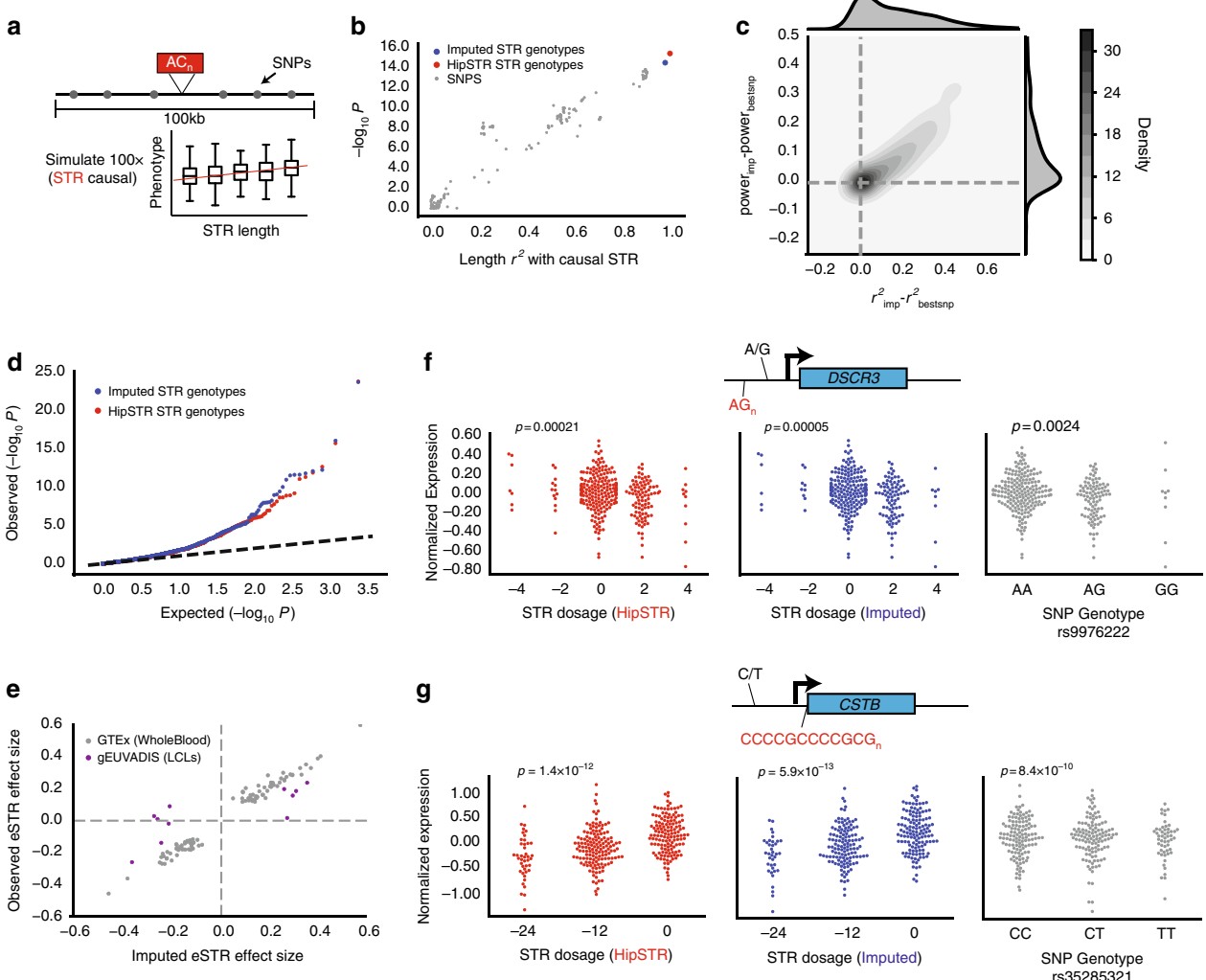

**Fig. 3** STR imputation improves power to detect STR associations. **a** Example simulated quantitative phenotype based on SSC genotypes. A quantitative phenotype was simulated assuming a causal STR (red). Power to detect the association was compared between the causal STR, imputed STR genotypes, and all common SNPs (MAF > 0.05) within a 50 kb window of the STR (gray). **b** Strength of association ($-\log_{10} p$) is linearly related with LD with the causal variant. For SNPs, the *x*-axis gives the length $r^2$ calculated using observed genotypes. For the imputed STR (blue), the *x*-axis gives the length $r^2$ from leave-one-out analysis. **c** The gain in power using imputed genotypes is linearly related to the gain in length $r^2$ compared to the best tag SNP. Gray contours give the bivariate kernel density estimate. Top and right gray area gives the distribution of points along the *x*- and *y*-axes, respectively. Power was calculated based on the number of simulations out of 100 with nominal $p < 0.05$. **d** Quantile-quantile plot for eSTR association tests. Each dot represents a single STR×gene test. The *x*-axis gives the expected $\log_{10}$ *p*-value distribution under a null model of no eSTR associations. Red and blue dots give $\log_{10}$ *p*-values for association tests using HipSTR genotypes and imputed STR genotypes, respectively. Black dashed line gives the diagonal. **e** Comparison of eSTR effect sizes using observed vs. imputed genotypes. Each dot represents a single STR×gene test. The *x*-axis gives effect sizes obtained using imputed genotypes. Gray dots give the effect size in GTEx whole blood using HipSTR genotypes. Purple dots give effect sizes reported previously[17] in lymphoblastoid cell lines. **f**, **g** Example putative causal eSTRs identified using imputed STR genotypes. Left, middle, and right plots give HipSTR STR dosage (red), imputed STR dosage (blue), and the best tag SNP genotype (gray) vs. normalized gene expression, respectively. STR dosage is defined as the average length difference from hg19. One dot represents one sample. *P*-values are obtained using linear regression of genotype vs. gene expression. STR and SNP sequence information is shown for the coding strand. Gene diagrams are not drawn to scale

We identified genes for which the STR is most likely the causal variant and tested whether STR imputation had greater power to identify causal eSTRs compared to SNP-based analyses. We used ANOVA model comparison to determine genes for which the STR explained additional variation over the top SNP (Methods). We additionally applied CAVIAR[47] to fine-map associations using the most strongly associated STR and the top 100 associated SNPs for each gene (Methods). We identified three genes with ANOVA $p < 0.05$ for which the STR was the top variant returned by CAVIAR. One example, a CG-rich STR in the promoter of *CSTB*, was previously demonstrated to act as an eSTR[48] and expansions of this repeat are implicated in myoclonus epilepsy[49].

In each case, imputed STR genotypes were more strongly associated with gene expression compared to the best tag SNP (Fig. 3f–g, Supplementary Table 4).

**Imputing normal alleles at known pathogenic STRs.** Finally, to determine whether alleles at known pathogenic STRs could be accurately imputed, we examined results of our imputation pipeline at 12 STRs previously implicated in expansion disorders that were included in our panel (Table 2). Our analysis focused on alleles in the normal repeat range for each STR, since pathogenic repeat expansions at these STRs are unlikely to be

**Table 2 Imputation performance at known pathogenic repeats**

| Locus | Motif | Disorder[a] | Length $r^2$ LOO | Observed concordance | Naive concordance | Random concordance | Best tag SNP | $r^2_{bestSNP}$ |
|---|---|---|---|---|---|---|---|---|
| 3:63898362 | CAG | SCA7 | 0.75 | 92.0% | 75.6% | 63.9% | rs58676857 | 0.57 |
| 4:3076604 | CAG | HD | 0.47 | 64.3% | 39.4% | 27.5% | rs762855 | 0.11 |
| 5:146258292 | CAG | SCA12 | 0.88 | 93.8% | 59.9% | 46.3% | rs2082405 | 0.64 |
| 6:16327867 | CAG | SCA1 | 0.72 | 85.3% | 55.0% | 33.8% | rs17860797 | 0.04 |
| 6:170870996 | CAG | SCA17 | 0.51 | 80.0% | 39.8% | 31.5% | rs9472489 | 0.15 |
| 12:112036755 | CAG | SCA2 | 0.49 | 96.2% | 88.2% | 80.2% | rs148019457 | 0.28 |
| 12:7045892 | CAG | DRPLA | 0.86 | 81.2% | 38.8% | 24.9% | rs34199021 | 0.69 |
| 13:70713516 | CTG/ CAG | SCA8 | 0.87 | 84.7% | 27.0% | 24.0% | rs9564660 | 0.39 |
| 14:92537355 | CAG | SCA3 | 0.88 | 86.4% | 33.8% | 27.5% | rs7144492 | 0.27 |
| 16:87637894 | CAG | HDL | 0.55 | 88.2% | 55.2% | 46.5% | rs2434850 | 0.34 |
| 19:46273463 | CTG | DM1 | 0.87 | 86.9% | 39.4% | 30.8% | rs7254351 | 0.44 |
| 19:13318673 | CAG | SCA6 | 0.81 | 92.0% | 44.1% | 39.2% | rs2070737 | 0.63 |

[a]HD Huntington's disease; SCA spinocerebellar ataxia, DRPLA Dentatorubral-pallidoluysian atrophy, DM1 myotonic dystrophy type 1, HDL huntington's disease-like 2. The best tag SNP for an STR is defined as the SNP within 50 kb with the highest length $r^2$. LOO refers to leave-one-out analysis in the SSC cohort $r^2_{bestSNP}$ gives the length $r^2$ between STR genotype length and the genotype of the best tagging SNP within 50 kb of the STR

present in the SSC cohort. Notably, accurate imputation of non-pathogenic allele ranges is still informative as (1) long normal or intermediate size alleles may result in mild symptoms in some expansion disorders[50–52] (2) longer alleles are more at risk for expansion[53] and (3) allele lengths below the pathogenic range could potentially be associated with more complex phenotypes[51].

Similar to the CODIS markers, these STRs are highly polymorphic with 10 or more alleles per locus. In all cases, imputed genotypes were more strongly correlated with observed genotypes compared to the best tag SNP. Where both HipSTR and Tredparse genotypes were available, concordance results were nearly identical across all STRs (Supplementary Table 5). Visualization of SNP-STR haplotypes at the CAG repeat implicated in dentatorubral-pallidoluysian atrophy (DRPLA)[54] reveals a typical complex relationship between STR allele length and local SNP haplotype (Fig. 4a), with the same STR allele often present on multiple SNP haplotype backgrounds. Still, for most STRs there is a clear association of specific haplotypes with different allele length ranges allowing accurate imputation across a large range of allele sizes (Fig. 4b, Supplementary Fig. 10).

Resolution of SNP-STR haplotypes can be used to infer the mutation history of a specific STR locus[26,27]. Notably, for many STR expansion orders it has been shown that pathogenic expansion alleles originated from a founder haplotype[55–58] associated with a long allele. We compared SNP haplotypes at the DRPLA locus in our dataset to a previously reported founder haplotype[55]. In concordance with the hypothesis of a single founder haplotype, we found that SNP haplotypes with smaller Hamming distance to the known founder haplotype had longer CAG tracts (Pearson $r = -0.79$; $p < 10^{-200}$). This finding demonstrates that while we were unable to directly impute pathogenic expansion alleles, STR imputation can accurately identify which individuals are at risk for carrying expansions or pre-pathogenic mutations and the inferred haplotypes can reveal the history by which such mutations arise.

## Discussion

Our study combines available whole-genome sequencing datasets with existing bioinformatics tools to generate the first phased SNP + STR haplotype panel allowing genome-wide imputation of STRs into SNP data. Despite their exceptionally high rates of polymorphism, 92% of STRs in our panel could be imputed with at least 90% concordance, and 38% achieved greater than 99% concordance. Imputation performance varied widely across STRs,

primarily due to differences in polymorphism levels across loci. Bi-allelic STRs could be imputed nearly perfectly (average concordance >99%, compared to 80% expected under a naive model), whereas STRs with the highest heterozygosity, including forensic markers and known pathogenic repeats, could be imputed to around 70% concordance (compared to approximately 50% expected under a naive model). We additionally show that imputation improves power to detect STR associations over standard SNP-based GWAS and could detect both known and previously unknown associations between STR lengths and expression of nearby genes.

A widely recognized limitation of GWAS is the fact that common SNP associations still explain only a small fraction of heritability of most traits. Multiple explanations for this have been proposed, including minute effect sizes of individual variants and a potential role for high-impact rare variation[59]. However, studies in large cohorts reaching hundreds of thousands of samples[1–3], as well as deep sequencing studies to detect rare variants[60], have so far not confirmed these hypotheses. An increasingly supported idea is that complex variants not well tagged by SNPs may comprise an important component of the missing heritability[10–12]. GWAS is essentially blind to contributions from highly polymorphic STRs and other repeats, despite their known importance to human disease and molecular phenotypes. Thus, STR association studies will undoubtedly uncover additional heritability that is so far unaccounted for. Notably, while autism phenotypes are available for the SSC families, this cohort is too small to perform a GWAS and was specifically ascertained for families enriched for de novo, rather than inherited, pathogenic mutations. In future work our panel can be applied to impute STRs into larger cohorts for autism and other complex traits for which tens of thousands of SNP array datasets are available.

Our initial haplotype panel faces several important limitations. First, the majority of samples are of European origin, limiting imputation accuracy in other population groups. Second, imputation accuracy is mediocre for the most highly polymorphic STRs, some of which will ultimately have to be directly genotyped to adequately test for associations. Notably, our work relied on existing tools originally designed for SNP imputation. Further work on computational methods specifically for imputing repeats may be able to improve performance. Finally, thousands of long STRs are filtered from our panel due to the limitation imposed by short read lengths. While we have included target STRs implicated in STR expansion disorders, many long STRs are still

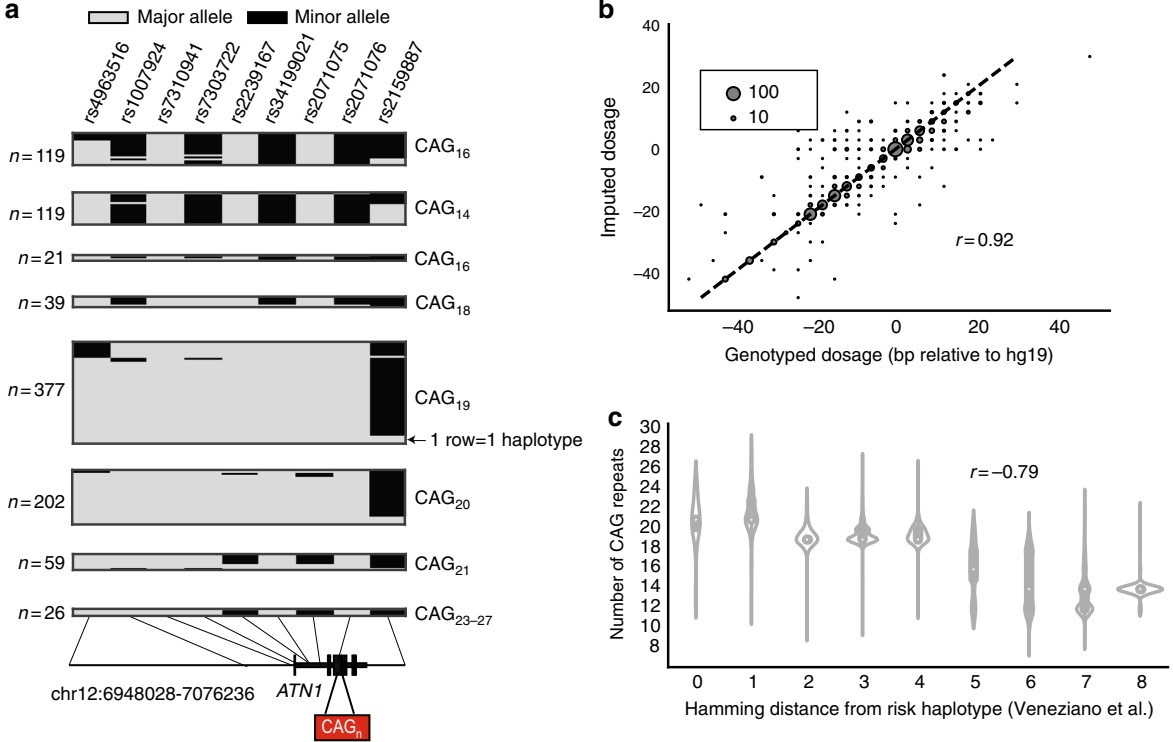

**Fig. 4** SNP haplotypes distinguish allele lengths at known pathogenic STRs. **a** Example SNP-STR haplotypes inferred in European samples at a polyglutamine repeat in *ATN1* implicated in DRPLA. Each column represents a SNP from the founder haplotype reported by Veneziano et al. Each row represents a single haplotype inferred in 1000 Genomes Project phase 3 European samples, with gray and black boxes denoting major and minor alleles, respectively. Haplotypes are grouped by the corresponding STR allele. The number of SNP haplotypes for each group of STR alleles is annotated to the left of each box. Alleles seen fewer than 10 times in 1000 Genomes samples were excluded from the visualization. **b** Comparison of imputed vs. observed STR genotypes in SSC samples at the DRPLA locus. The *x*-axis gives the maximum likelihood genotype dosage returned by HipSTR and the *y*-axis gives the imputed dosage. Dosage is defined as the sum of the two allele lengths of each genotype relative to the hg19 reference genome. The bubble size represents the number of samples summarized by each data point. **c** Distribution of DRPLA repeat length vs. similarity to the pathogenic founder haplotype. The founder haplotype refers to the SNP haplotype reported by Veneziano, et al. on which a pathogenic expansion in *ATN1* implicated in DRPLA likely originated. The *x*-axis gives the Hamming distance between observed haplotypes and the founder haplotype, computed as the number of positions with discordant alleles. White dots represent the median length

inaccessible using current tools. New methods are now being developed for genome-wide genotyping of more complex STRs[37,61] and longer variable number tandem repeats (VNTRs)[62] from short reads and can be used to expand our panel in the future. Overall, our STR imputation framework will enable an entire new class of variation to be interrogated by reanalyzing hundreds of thousands of existing datasets, with the potential to lead to novel genetic discoveries across a broad range of phenotypes.

## Methods

**SSC Dataset**. The SSC Phase 1 dataset consists of 1916 individuals from 479 quad families. Access to SSC data was approved for this project under SFARI Base project ID 2405.1. This study was certified as exempt from institutional review board (IRB) review by the University of California San Diego IRB (Project #161286XX) since only de-identified data was accessed. Informed consents were obtained for each participating family by SSC recruitment sites in accordance with their local IRBs.

Aligned BAM and gVCF files for whole-genome sequencing data of individuals were obtained through SFARI base (see URLs) and processed on Amazon Web Services (AWS). SNP genotypes were called from gVCF files using the GATK version 3 joint calling pipeline[63]. A total of 27,185,239 variants that passed the default GATK filters and overlapped with sites reported in the 1000 Genomes Project[38] phase 3 data were retained for downstream analysis.

We performed principal components analysis (PCA) using SNPs from 2504 samples from Phase 3 of the 1000 Genomes Project[38] and projected SSC samples onto the resulting PCs to infer sample ancestry (Supplementary Fig. 1). We estimated that the SSC cohort consists of 1585 Europeans, 39 East Asian, 172 South

Asian, 69 African samples, and 51 individuals that did not clearly belong to any single population group.

**Genome-wide multi-sample STR genotyping**. STRs were jointly genotyped on the AWS EC2 platform in batches of 500 STRs. We streamed the corresponding region of each BAM file and of the phased SNP VCF files to a local EBS volume attached to each EC2 instance using samtools[64] version 1.4 and tabix[65] version 1.2, respectively. HipSTR[32] version v0.5 was called individually per STR with default parameters. Phased SNPs were provided as input to allow HipSTR to perform physical phasing when possible. Resulting VCF files from each batch were merged to create a genome-wide callset in VCF format.

HipSTR calls were filtered using the filter_vcf.py script in the HipSTR package with suggested parameters (--min-call-qual 0.9 --max-call-flank-indel 0.15 --max-call-stutter 0.15). We used the following criteria to remove problematic STRs from the callset: (i) STRs overlapping segmental duplications (UCSC Table Browser[66] hg19.genomicSuperDups table) were removed from the callset using intersectBed[67] v2.25.0; (ii) Pentanucleotides and hexanucleotides containing homopolymer runs of at least 5 or 6 nucleotides, respectively, in the hg19 reference genome were removed as they were found to contain an excess of indels in the homopolymer regions; (iii) STRs with call rate <80%; (iv) STRs with expected heterozygosity <0.095, corresponding to a minor allele frequency of 5% for bi-allelic markers, were removed to restrict to polymorphic STRs; (v) STRs with significantly more or fewer heterozygous genotypes compared to expectation under Hardy-Weinberg equilibrium ($p < 0.01$) as suggested previously[68]. After filtering, 453,671 STRs remained in our panel.

**Genotyping clinically relevant STRs**. A total of 25 clinically relevant STRs were called using Tredparse[37] v0.75 from the aligned BAM files obtained through SFARI base on Amazon EC2. Default profiles containing information about the genomic position, reference repeat length, and repeat motif supplied with the software were

used. We filtered STRs with call rate less than 80% or for which only a single allele was identified (Supplementary Table 1). Nine STRs remained after filtering.

**Computing expected STR heterozygosity.** For an STR with alleles $\{1...n\}$, let $p_i$ be the frequency of the $i$th allele computed from observed genotypes. Expected STR heterozygosity is defined as: $H = 1 - \sum_{i=1}^{n} p_i^2$. For this study all alleles with identical length are treated as the same allele. On average each length-based allele corresponded to 1.8 sequence-based alleles.

**Comparison to 1000G catalog.** STR genotypes for 1000 Genomes samples generated by Willems et al.[14] were downloaded from the strcat site (see URLs). Expected heterozygosity was computed using the PyVCF package (see URLs) for the 1000 Genomes calls and using a custom script for the SSC data to collapse alleles of identical length into a single allele. STRs passing all filters described above included in the comparison. Analysis was restricted to STRs with at least 500 calls in the 1000 Genomes dataset.

**Normal allele frequency distributions at pathogenic STRs.** Control distributions for Fig. 1e were obtained from previous studies of normal alleles at known pathogenic STRs. Allele frequencies for SCA1, SCA2, SCA3, SCA6, SCA12, SCA8, SCA17, and DRPLA were obtained from Fig. 1 of Majounie et al.[36] and are based on 307 controls of Welsh origin. Frequencies for DM1 were obtained from Fig. 1 of Ambrose et al.[35] and are based on 254 controls of Chinese origin. Frequencies for HDL were obtained from Fig. 1 of Figley et al.[34] and are based on 352 controls of North American Caucasian origin. Frequencies for SCA7 were obtained from Fig. 1 of Gouw et al.[33] and are based on 180 controls of European origin. Frequencies for HTT are based on data in the phv00173896.v1.p1 variable of dbGaP study phs000371.v1.p1 (Genetic modifiers of Huntington's Disease) based on the shorter allele of 2802 patients with Huntington's Disease.

**Phasing SNPs in the SSC.** SNP genotypes were phased using SHAPEIT[69] version 2.r837 with 1000 Genomes Phase 3 genotypes as a reference panel and ignoring pedigree information. SHAPEIT's duoHMM[70] version 0.1.7 method was used to refine phased haplotypes using pedigree structure and correcting for Mendelian errors.

**Phasing STRs.** Beagle[40] version 4.0 was used to phase each STR separately using phased SNP genotypes, pedigree information, and unphased STR genotypes as input. In order to leverage the HipSTR genotype likelihoods (GL field), Beagle requires all samples to have GL information. To accommodate this, phasing was performed in two steps. First, samples with missing data were removed and the remaining samples were phased using the -gl Beagle flag. Next, missing samples were added back to the VCF and all samples were jointly phased in a second Beagle round using default parameters. In this step Beagle additionally imputed any calls with missing genotypes. Genotype values (GT field) were used for the STRs genotyped using Tredparse as it does not report genotype likelihoods, and phasing and imputation of STRs was done in a single step. Phased STRs and SNPs for only the unrelated parent samples from each locus were then merged into a single genome-wide reference panel in VCF format.

**Imputation performance metrics.** Let $X = \{x_1, x_2,...x_n\}$ be the true STR genotypes for samples $1..n$ and $Y = \{y_1, y_2,...y_n\}$ be the imputed STR genotypes. Each genotype $x_i$ is defined as $x_i = (x_{i1}, x_{i2})$ where $x_{i1}$ and $x_{i2}$ give the (unordered) lengths of the two STR alleles for a diploid sample and similarly for $Y$. We then define the following metrics:

Genotype concordance $c_i$ was defined as: 1 if both genotypes match ($x_{i1} = y_{i1}$ and $x_{i2} = y_{i2}$ or $x_{i2} = y_{i1}$ and $x_{i1} = y_{i2}$); 0 if neither imputed allele matched a true allele; else 0.5 if one but not both imputed alleles matched the true alleles. Genotype concordance for an STR is the average over all the samples $C = \frac{1}{n}\sum_{i=1}^{n} c_i$.

Define the STR genotype dosage as the sum of the lengths of the two alleles at a given site: $d_i = x_{i1} + x_{i2}$ and $X_d = \{d_1, d_2,...,d_n\}$. Length $r^2$ is computed as

$$\mathrm{cov}^2(X_d, Y_d)/(\mathrm{Var}(X_d)\mathrm{Var}(Y_d)).$$

For a given allele length $a$, define $X_a = \{a_1, a_2,...,a_n\}$ where $a_i = \sum_{j=1}^{2} 1_{(x_{ij}=a)}$. Allelic $r^2$ is computed as $\mathrm{cov}^2(X_a, Y_a)/(\mathrm{Var}(X_a)\mathrm{Var}(Y_a))$.

The best tag SNP for an STR is defined as the SNP within 50 kb with the highest length $r^2$.

For all concordance metrics, outlier genotypes containing alleles seen less than three times in the entire cohort were removed from the analysis.

For each STR, we additionally computed the expected value of each metric under a random model where genotypes are imputed randomly based on the frequency of underlying alleles and a naive model where genotypes are imputed to be the most common diploid genotype. Expected genotype concordance under the random model was calculated as $\sum_{i,j} f_i f_j \left(\sum_{k,l} C(i,j,k,l)\right)$, where $(i, j) \in \{1,...,n\}^2$ and $(k, l) \in \{1,...,n\}^2$, $n$ is the number of alleles, $f_x$ gives the frequency of allele $x$, and $C(i, j, k, l)$ gives the concordance between genotypes $(i, j)$ and $(k, l)$ as defined

above. For example, for a bi-allelic marker with allele frequencies $f_1$ and $f_2$ expected genotype concordance under the random model is given by $f_1^2(f_1^2 + (0.5)2(f_1)(f_2)) + 2f_1f_2((0.5)f_1^2 + 2f_1f_2 + (0.5)f_2^2) + f_2^2(f_2^2 + (0.5)2f_1f_2)$. Random model values for length $r^2$ and allelic $r^2$ were computed by comparing genotypes imputed randomly based on population allele frequencies to true genotypes at each STR. Concordance under the naive model was computed by comparing each sample's genotype to the most frequent diploid genotype. Length $r^2$ and allelic $r^2$ are not defined under the naive model since all imputed genotypes are identical.

**Evaluating imputation performance in the 1000 Genomes data.** STRs were imputed into SNP data downloaded from the 1000 Genomes Project site from three sources (WGS, phased SNPs from Affy6.0 array; and phased SNPs from Omni2.5 array; see URLs and Supplementary Table 3) with Beagle version 4.1 using the SSC SNP-STR haplotype panel. For comparison to WGS, STRs were jointly genotyped in high-coverage WGS datasets for 150 of the 1000 Genomes Project samples (see URLs) using HipSTR version 0.6 followed by the filtering steps described above for the SSC cohort.

Capillary electrophoresis genotypes for 209 samples at 721 Marshfield STRs were downloaded from the Payseur Lab website (see URLs). PCR product sizes were converted to length differences in bp from the reference genome using product size annotations[71] available from the Rosenberg Lab website (see URLs). Prior to comparing genotypes, offsets were calculated to match HipSTR lengths to the length of Marshfield STRs as previously described[14]. STRs with imperfect repeat structures were removed. Capillary genotypes were rounded down to the nearest number of repeat units.

10X Genomics data for NA12878 was obtained from the NA12878 Gemline Genome v2 available on the 10X Genomics website (see URLs). We extracted reads belonging to phase 1 or 2 from the phased, barcoded BAM based on the HP tag into separate BAM files. HipSTR v0.6.1 was called separately on each BAM with non-default parameters --def-stutter-model --min-reads 5 --use-unpaired and with --haploid-chrs containing a list of all autosomal chromosomes to force a haploid genotyping model. Haploid STR calls were obtained for both phases at 118,353 STRs. We identified the nearest heterozygous SNP to each STR that was genotyped in both the 10X data and in our phased panel. STRs for which the nearest SNP had discordant genotypes in the two datasets were discarded leaving 116,764 STRs for analysis.

**Simulations for power analysis.** We analyzed parental genotypes for 5838 STRs across chromosome 21 that passed filtering and quality control as described above. For each STR, we simulated quantitative phenotype datasets under the model: $P = \beta G + E$, where $P$ is a vector of standard normalized phenotypes, $\beta$ gives the effect size, $E$ gives the error term drawn from a normal distribution $N(0,1-\beta)$, and $G$ is a vector of the sum of genotype lengths for each individual scaled to have mean 0 and variance 1. For each simulated phenotype dataset, we tested the causal STR, the imputed STR genotypes, and the best tag SNP (strongest length $r^2$) within 50 kb of the STR for association. Association tests were performed using the Python statsmodels library OLS method (see URLs).

We performed additional simulations under a case–control model shown in Supplementary Fig. 8. Phenotypes (0 = control, 1 = case) were drawn for each sample according to the model logit $(p_i) = \beta X_i$ where $p_i$ is the probability that sample $i$ is a case and $X_i$ is the scaled genotype for individual $i$ as described above. Association tests were performed using the Python statsmodels Logit method.

For the non-additive phenotype example (Supplementary Fig. 9), we performed simulations under a quadratic model: $P = \beta G^2 + E$ where $G$ is a vector of the squared sum of allele lengths scaled by the mean allele length, and $P$, $\beta$, $E$ are as described above. Two sets of association tests were performed: the first tested for association between STR length and phenotype (Supplementary Fig. 9b) and the second set performed a separate association test for each STR allele treating the allele as a bi-allelic locus (Supplementary Fig. 9c).

In all cases 100 separate simulations were performed and power was defined as the percent of simulations for which the nominal association $p$-value was <0.05. Figures show results for all simulations with $\beta$ set to 0.1.

**eSTR analysis.** Data for eSTR analysis was obtained from the Genotype-Tissue Expression (GTEx) through dbGaP under phs000424.v7.p2. This included high-coverage (30×) Illumina whole-genome sequencing (WGS) data from 650 unrelated samples, Omni 2.5 SNP genotypes for 450 samples, and gene-level RPKM values for whole blood in 336 samples. STRs were genotyped from WGS data using HipSTR v0.5 and subject to the same quality filtering as SSC samples. STRs were additionally imputed to Omni2.5 data with Beagle as described above. Downstream analyses were restricted to the 336 samples with available whole blood expression data. These samples consisted of 284 European, 45 African American, 3 Asian, and 3 Amerindian samples and 2 samples with no population label available.

We performed separate eSTR analyses using HipSTR and imputed genotypes. In each case, we performed a separate association test between gene expression and each STR within 100 kb of the gene using a model $Y = \beta X + C + \varepsilon$, where $X$ denotes STR genotype lengths, $Y$ denotes expression values, $\beta$ denotes the effect size, $C$ denotes various covariates, and $\varepsilon$ is the error term. Following our previous study[17], we used STR dosage, defined as the sum of repeat lengths of the two alleles

for each sample, to define STR genotypes. All repeat lengths are reported as length differences from the hg19 reference, with 0 representing the reference allele. STR dosages were scaled to have mean 0 and variance 1. Genes with median expression of 0 were excluded and expression values for remaining genes were quantile normalized to a standard normal distribution. We included sex, population structure, and technical variation in expression as covariates. For population structure, we used the top 15 principal components resulting from perform principal components analysis on the matrix of SNP genotypes from each sample. To control for technical variation in expression, we applied PEER factor correction[72,73] using 83 PEER factors.

We used model comparison to determine whether the best eSTR for each gene explained variation in gene expression beyond a model consisting of the best eSNP. For each gene with an eSTR we determined the lead eSNP with the strongest $p$-value. We then compared two linear models: $Y \sim$ eSNP (SNP-only model) vs. $Y \sim$ eSNP + eSTR (SNP + STR model) using the anova_lm function in the python statsmodels.api.stats module. We used CAVIAR v1.0 to further fine-map eSTR signals against the top 100 eSNPs within 100 kb of each gene. Pairwise-LD between the eSTR and eSNPs was estimated using the Pearson correlation between SNP dosages (0, 1, or 2) and STR dosages (sum of the two repeat allele lengths).

**Comparison to DRPLA founder haplotypes**. The founder haplotype for the expansion allele in *ATN1* implicated in DRPLA was taken from Table 1 of Veneziano et al.[55] and consists of rs4963516, rs1007924, rs7310941, rs7303722, rs2239167, rs34199021, rs2071075, rs2071076, and rs2159887 with hg19 alleles G, A, G, T, A, A, T, C, and C, respectively. Distance from the founder haplotype was calculated as the number of mismatches.

**URLs**. For Simons Simplex Collection, see https://base.sfari.org/. For HipSTR, see https://github.com/tfwillems/HipSTR. For Beagle, see https://faculty.washington.edu/browning/beagle/b4_0.html. For 1000 Genomes phased Affy6.0 and Omni2.5 SNP data, see ftps.1000genomes.ebi.ac.uk/vol1/ftp/release/20130502/supporting/shapeit2_scaffolds/hd_chip_scaffolds/. For 1000 Genomes Phase 3, see http://ftp.1000genomes.ebi.ac.uk/vol1/ftp/release/20130502/. For 1000 Genomes STR data, see http://strcat.teamerlich.org/download. For Marshfield Capillary electrophoresis data, see https://payseur.genetics.wisc.edu/strpData.htm. For Marshfield marker annotations, see https://web.stanford.edu/group/rosenberglab/data/pembertonEtAl2009/Pemberton_AdditionalFile1_11242009.txt. For NA12878 10X Genomics data, see https://support.10xgenomics.com/genome-exome/datasets/2.2.1/NA12878_WGS_v2. For High-coverage Illumina sequencing for 1000 Genomes samples, see https://www.ebi.ac.uk/ena/data/view/PRJEB20654. For PyVCF, see https://github.com/jamescasbon/PyVCF. For Python statsmodels, see http://www.statsmodels.org/stable/index.html.

**Code availability**. Analysis scripts and Jupyter notebooks for reproducing the figures in this study are provided in the Github repository https://github.com/gymreklab/snpstr-imputation.

## Data availability

Phased SNP-STR haplotypes for 1000 Genomes Project phase 3 samples and example commands for imputation are available from Gymrek Laboratory webpage [https://gymreklab.github.io/2018/03/05/snpstr_imputation.html]. Phased SNP-STR haplotypes for the SSC samples are available through SFARI base Accession Code: SFARI_SSC_WGS_1c. 1000 Genomes phased Affy6.0 and Omni2.5 SNP data are available through the 1000 Genomes FTP server [ftp.1000genomes.ebi.ac.uk/vol1/ftp/release/20130502/supporting/shapeit2_scaffolds/hd_chip_scaffolds/]. 1000 Genomes phase 3 Whole-Genome Sequencing data is available through the 1000 Genomes FTP server [http://ftp.1000genomes.ebi.ac.uk/vol1/ftp/release/20130502/]. 1000 Genomes STR data is available from strcat [http://strcat.teamerlich.org/download]. Marshfield Capillary electrophoresis data is available from the Payseur Laboratory webpage [https://payseur.genetics.wisc.edu/strpData.htm]. Marshfield marker annotations are available from the Rosenberg Laboratory webpage [https://web.stanford.edu/group/rosenberglab/data/pembertonEtAl2009/Pemberton_AdditionalFile1_11242009.txt]. NA12878 10X Genomics data is available at the 10X Genomics Datasets Repository [https://support.10xgenomics.com/genome-exome/datasets/2.2.1/NA12878_WGS_v2]. High-coverage Illumina sequencing for 1000 Genomes samples is available from the European Nucleotide Archive Accession Code PRJEB20654

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

## Acknowledgements

Research reported in this publication was supported in part by the Office Of The Director, National Institutes of Health under Award Number DP5OD024577 and by a SFARI Explorer Award Number 515568. M.G. was supported in part by NIH/NIMH grant R01 MH113715. This work used the Extreme Science and Engineering Discovery Environment (XSEDE) comet resource at the San Diego Supercomputing Center through allocations ddp268 and csd568. XSEDE is supported by National Science Foundation grant number ACI-1548562. We thank Alon Goren for helpful comments on the manuscript. We additionally thank Vineet Bafna and Vikas Bansal for helpful discussions and providing access to compute resources. We are grateful to all of the families that participated in the Simons Simplex Collection as well as the principal investigators.

## Author contributions

M.G. conceived the study, helped design and perform analyses, and drafted the initial manuscript. S.S. generated the reference haplotype panel, performed downstream analyses, and participated in writing the manuscript. I.M. performed simulation analyses. S. F.F. performed analyses of expression data. N.M. performed analyses of pathogenic STRs. All authors have read and approved the final manuscript.

## Additional information

**Competing interests:** The authors declare no competing interests.

