## [Peer Review File · Nature Communications]

Reviewer #1 (Remarks to the Author):

This manuscript describes a novel approach allowing the imputation of short tandem repeat (STR) genotypes via single nucleotide polymorphism (SNP) array data. STRs have been largely overlooked in genetics studies, particularly genome-wide association studies (GWAS) in humans, and the reference haplotype panel for genome-wide imputation of STRs has the potential to open up new data-mining of large existing data-sets. The approach has potentially broad utility and would be of interest to many in diverse fields of human/clinical genetics, as well as the genetics of many other species (for example for agricultural and ecological/environmental applications). The data is clearly presented and the manuscript is generally well written. Importantly, the authors also discuss the limitations of their approach. I have some suggestions below as to how the manuscript could be improved.

1. Considering that there are over 40 tandem repeat disorders, Table 2 seems to analyse a fairly limited subset of these disorders. Could it be expanded? Covering at least all of the CAG/glutamine expansion disorders would be more systematic. However other disorders and STRs such as fragile X syndrome (and the causative trinucleotide repeat in the FMR1 gene) would be interesting. There are various other tandem repeat disorders, including Friedreich ataxia and the C9orf72 hexanucleotide expansion neurodegenerative diseases, that would also be of interest.
2. Along the same lines, it would be of great interest to validate this novel imputation approach, and the associated reference haplotype panel, for a major complex disorder in which large-scale GWAS has been published, such as schizophrenia.
3. For the information being conveyed in Figure 4A (e.g. if 1 row = 1 haplotype, how many haplotypes are included in each box?) the figure legend could be made more clear, to make it more accessible to a broad readership.
4. One minor formatting issue is that in Figure 2A, the grey and black letters can be difficult to distinguish – I suggest grey letters be re-formatted to increase the contrast from the black (and red) letters. Also, in the Figure 2 legend, hyphens should be added to read 'leave-one-out analysis'. In the Figure 3 legend, a hyphen should be removed to read 'leave-one-out analysis'.
5. One of the major implications of this manuscript is that it may help establish whether STRs contribute to 'missing heritability' for complex disorders and traits. In the Discussion, Reference 10 is the earliest cited article purporting that tandem repeats might be contributors to 'missing heritability', however this idea had been proposed earlier (Trends Genet. 2010; 26:59-65).

Reviewer #2 (Remarks to the Author):

Saini et al carried out SNP-STR phasing and impute the genotypes at STR loci. The experiments show that the procedure may increase the value of GWAS data. Some specific comments are below:

1. The concordance rate has suggested overall good performance of the pipeline. However, the concordance rate can appear more optimistic than they really is, particularly when sites are lumped together in this fashion. First of all, the concordance is highly affected by heterozygosity (which is indeed addressed in Figure 2B). It may be a good practice to remind the readers what the concordance rate would be under a random model. For example, for a bi-allelic site with allele frequencies p and q . The expected concordance would be $p^2 + q^2$ ($p^2 \cdot 1 + 2pq \cdot 0.5 + 2pq \cdot 0.5 + q^2 \cdot 1$). The authors used a cutoff of 5% allele frequency (heterozygosity of 0.095), which still contain lots of sites that will naturally give a high concordance rate. For example, when $p = 0.95$, the concordance would be 91% expected by random; when $p = 0.9$, the concordance rate would be 84%; when $p = 0.5$, the concordance would be 63%. So, it would be nice to show a distribution of heterozygosity for all

STR sites in the paper. Then, for each heterozygosity level, indicate the expected concordance rate under a random model. This way, readers might get a better idea on how the overall concordance is comprised of, instead of lumped together. Specifically, in the Figures (Figure 2BC) and tables (Table 1 and 2), indicate where the 'random' concordance is, as a baseline to compare to.

2. Another factor which the authors could indicate in their analyses is the base pair distance from the STR site to the best tagging SNP site. Then correlate or stratify the distance in respect to concordance rate, as well as length r^2 and biallelic r^2 . This helps to get a picture on what sort of haplotype structure is expected in the phasing and imputation, and how that may affect the accuracy.

3. The section that claimed to show better GWAS utility is weak. It is not much of a surprise that the causal/imputed STR offers better power, since they are part of the simulation itself. Are there public phenotype/expression datasets with known associations that can be tested here and see if one can reproduce here?

4. What happens to the famous Huntington's disease locus, that should also contain 'normal' alleles in the range assayable by the imputation method? However, it is not included in Table 2.

Critical comments:

1. Probably the weakest point of the study is the lack of a true truth set, and thus the reliance on various levels of approximation: phasing is based on statistical approaches (Beagle), as it is the case also for STRs (HipSTR, LobSTR). Much of the experimentation uses simulation rather than actual physical data, or disease alleles. There is no post-hoc validation of some critical sites for the accuracy of imputation from genotyping data.

2. As the authors openly state, they filtered long STRs, and they had limited accuracy with heterozygosis. These are important consideration to its applicability for human disease.

3. In some way, this approach may suffer of the same limitations of tools for the prediction of structural variation from GWAS data.

In summary, this is a good addition to the toolkit for existing GWAS studies. However, it may be suboptimally tested for actual accuracy. The value of this work would increase significantly if there were, at least, an effort at post-hoc assessment of the precision of the tool for medically relevant loci.

Reviewer #3 (Remarks to the Author):

A reference haplotype panel for genome-wide imputation of short tandem repeats

This study uses genomic data from families to produce a set of reference haplotypes involving both SNPs and STRs. The authors run Beagle to impute STRs in a genome-wide dataset, obtaining a high rate of concordance between imputed and known genotypes. The paper is a milestone in human STR studies, which have long been interested in LD between STRs and SNPs. The study does for STRs much of what early haplotype map and imputation studies did for SNPs.

One issue that could be better explored is the basis for the variation in imputation accuracy. From SNP studies, the accuracy is expected to depend on the level of LD between the SNPs and STRs, the level of variation of the STRs, and the population of origin of the samples. The authors could dig through the results a bit more to report on the impact of these various factors on the outcomes. The most difficult loci have moderate accuracy well below the 97% result in the abstract. The comparison to the CODIS imputation of Edge et al. 2017 is interesting in this regard. With their larger sample, the pedigree information, the greater population homogeneity, and the more favorable leave-one-out approach, the authors' imputation accuracies at the CODIS loci are

not much higher than in Edge et al. The study is possibly near the limit of what can be achieved for some of the most highly polymorphic loci.

A second comment is that the study could reach farther into the literature to make its significance clear as more than a technical advance to GWAS with STRs. Some studies along the way toward this imputation work include the early study of LD between an STR and a SNP in Tishkoff et al. 1996 Science and the SNPSTRs of Mountain et al. 2002 Genome Res. A key reference is Payseur et al. 2008 AJHG, which certainly merits some discussion as background.

Although the test the authors use for power to detect association with the repeat length of an STR is nice, they could magnify the impact by choosing a simulated trait that is connected to an STR in a more indirect way. This would evaluate the benefit of STR imputation for a more typical trait.

Abstract

Capture on average 20% more variation... compared to individual common SNPs – the exact computation and meaning of the statement are not clear here (see also p. 4).

Highlighting a limitation – the limitation is that STRs are not typically studied.

Introduction

One of the largest sources of variation – clarify the sense in which this is meant.

Genotyping errors introduced during PCR amplification – this topic goes back to early STR studies (see e.g. Lai et al. 2003 J Comp Biol and Lai & Sun 2003 J Theor Biol).

Results

Mostly European – give the population composition with counts here.

Profile autosomal STRs – clarify what HipSTR does (in particular, explain the meaning of “profile”).

“loci” on p. 5 – better to use “STR” or “SNP” to avoid ambiguity.

Heterozygosities – how are these computed? Are these observed heterozygosities or expected heterozygosities? This is ambiguous through the paper.

Which is not supported – this is a little awkward, I think you mean that Beagle is the only program that accommodates all the features in the sentence.

Consistent with observations from SNP imputation – it is not clear this is entirely consistent. East Asians have slightly higher LD compared to Europeans. So with all other factors being equal (e.g. size of reference panel), East Asians will be expected to have higher concordance.

Only long alleles have pathogenic effects – add reference.

Resolution of SNP-STR haplotypes – this type of joint SNP-STR analysis was common in earlier SNP-STR studies (e.g. Tishkoff et al. 1996 Science, Mountain et al. 2002 Genome Res, Schroeder et al. 2009 MBE).

Discussion

In this first paragraph it would be good to give some numbers, e.g. the fraction of loci where specific concordance thresholds are achieved.

Methods

Open with a description of exactly what the data are – how many samples from each of the populations, how many SNPs and STRs were detected, etc.

Collapse alleles of identical length into a single allele – describe how often this heterogeneity is collapsed.

Figures

Some of these plots could be presented using a more informative heat map e.g. Figs 1D, 2D, 2E, S5. Fig 7B, 8B, and 8C also could be presented without so many overlapping points.

Fig S1. In the caption, describe the number of individuals in each group.

Fig S2. This plot is likely to be partially explained by the fact that r^2 is bounded above by a function of the allele frequencies at a pair of loci. The high polymorphism at the STRs is likely to lead to allele frequencies that differ between the STR and the SNP, so that the bound on r^2 is lower.

Fig S3. Make this caption more self contained, describing the method and the meaning of "concordance."

Fig S6. Explain how the best tag SNP is found. What is the meaning of "alternate"?

A reference haplotype panel for genome-wide imputation of short tandem repeats

Reply to reviewers

Overview

We thank the reviewers for their careful reading of the manuscript and insightful comments. Overall, we were encouraged by the reviewers' enthusiasm and description of this work as "a milestone in human STR studies" and "of broad utility and of interest to many in diverse fields of human/clinical genetics." They raised several points and suggestions, which we address in full in the revised manuscript. We highlight here the major points addressed in our revision:

Inclusion of clinically relevant repeat expansion loci: Our haplotype panel has been updated to include known pathogenic repeats, including the polyglutamine repeats implicated in Huntington's Disease and hereditary ataxias (See **Table 2**).

Validation of inferred genotypes and haplotypes: We now include validation against STR genotypes obtained using capillary electrophoresis for the highly polymorphic Marshfield STRs (**Figure 2D**) and genome-wide phased STR genotypes obtained using HipSTR on data for NA12878 from 10X Genomics, a synthetic long read technology (**Figure 2E**). In all cases, per-STR concordance results matched closely to those estimated based on HipSTR calls.

Application to discovering novel associations: We and others^{1,2} have recently demonstrated that STRs likely contribute to regulating expression of nearby genes (termed "eSTRs"). We now demonstrate the ability of imputed STRs to identify known and novel phenotype associations using RNA-sequencing and WGS available from the Genotype-Tissue Expression (GTEx) Project (**Figure 3D-G**).

In addition to the major areas addressed above, we have answered each comment in depth below. Reviewer comments are given in **blue text**, and our responses are in black. When relevant, we additionally provide the page number where each change can be found in the main text in **bold red**. Finally, all changes to the main text are indicated there in **red font**.

Reviewer #1 (Remarks to the Author):

This manuscript describes a novel approach allowing the imputation of short tandem repeat (STR) genotypes via single nucleotide polymorphism (SNP) array data. STRs have been largely overlooked in genetics studies, particularly genome-wide association studies (GWAS) in humans, and the reference haplotype panel for genome-wide imputation of STRs has the potential to open up new data-mining of large existing data-sets. The approach has potentially broad utility and would be of interest to many in diverse fields of human/clinical genetics, as well as the genetics of many other species (for example for agricultural and ecological/environmental applications). The data is clearly presented and the manuscript is generally well written.

Importantly, the authors also discuss the limitations of their approach. I have some suggestions below as to how the manuscript could be improved.

We thank the reviewer for the positive feedback. We have addressed each comment in detail below.

1. Considering that there are over 40 tandem repeat disorders, Table 2 seems to analyse a fairly limited subset of these disorders. Could it be expanded? Covering at least all of the CAG/glutamine expansion disorders would be more systematic. However other disorders and STRs such as fragile X syndrome (and the causative trinucleotide repeat in the FMR1 gene) would be interesting. There are various other tandem repeat disorders, including Friedreich ataxia and the C9orf72 hexanucleotide expansion neurodegenerative diseases, that would also be of interest.

As mentioned above in “**Inclusion of clinically relevant repeat expansion loci**”, we have expanded our panel to include additional pathogenic repeats, including the Huntington’s Disease locus and many of the spinocerebellar ataxias. **Table 2** now includes 12 known pathogenic repeats in total. (p. 18)

Our original panel is based on HipSTR genotypes, which are limited to repeats that can be spanned by short reads. However, even normal range alleles for most STR expansion disorders are close to or beyond Illumina read lengths, and thus do not pass HipSTR quality filters. To supplement our panel, we have additionally genotyped target pathogenic loci using the recently developed Tredparse tool, which can genotype repeats that are longer than the read length. We achieve average concordance of 86% between imputed and observed genotypes at known pathogenic STRs. For STRs called by both Tredparse and HipSTR, genotype calls and imputation quality metrics were nearly identical across methods (**Supplementary Tables 1, 7**).

Notably, since our panel focuses on autosomal loci, we have not included the Fragile X locus. Additionally, while the Friedreich ataxia and C9orf72 loci were included in the Tredparse panel, no calls were returned and thus these were excluded from our panel. The ability to genotype longer repeats from short reads is fast-evolving. As we and others improve algorithms for genome-wide analysis of long repeats³⁻⁵, we envision further supplementing our panel in the future.

2. Along the same lines, it would be of great interest to validate this novel imputation approach, and the associated reference haplotype panel, for a major complex disorder in which large-scale GWAS has been published, such as schizophrenia.

As described above in “**Application to discovering novel associations**”, we have included a validation of our approach by demonstrating the ability to detect associations of STRs with gene expression using imputed genotypes from the GTEx cohort.

We tested for association between STR length and expression of nearby genes in whole blood using data from chromosome 21. Association results based on imputed genotypes were highly similar to those based on HipSTR genotypes ($r=0.99$; $p=1.01 \times 10^{-79}$, $n=97$) and to effect sizes we previously reported using lymphoblastoid cell lines from an orthogonal set of samples ($r=0.79$; $p=0.0042$, $n=11$). We used a fine-mapping approach to identify STRs like to be causal variants, including a previously known eSTR for *CSTB*⁶ implicated in myoclonus epilepsy^{6,7}. We found that imputed STRs captured these associations far better than tagging SNPs (**Figure 3F, G**). (pp. 9-10)

Notably, analysis here has been restricted to a single chromosome and tissue because we are in the process of preparing a separate manuscript on genome-wide eSTRs across tissues in the GTEx cohort. Our results in that study indicate hundreds of novel eSTRs which we plan to describe in detail there. We believe the current analyses we have added to this study provide a proof of principle that STR imputation can identify novel associations with higher power than SNP-based studies and with similar power to STR genotypes obtained from WGS. In future work we and others aim to perform large-scale association studies with STRs using available GWAS datasets, such as the PGC cohort for schizophrenia. However as we have discussed with the editor we determined that identification of novel disease loci is beyond the scope of the current manuscript.

3. For the information being conveyed in Figure 4A (e.g. if 1 row = 1 haplotype, how many haplotypes are included in each box?) the figure legend could be made more clear, to make it more accessible to a broad readership.

Figure 4A now displays the number of haplotypes included in each box. The **Figure 4** legend has been expanded to be more self-contained and define terms that may not be clear to a broad readership. (p. 16)

4. One minor formatting issue is that in Figure 2A, the grey and black letters can be difficult to distinguish – I suggest grey letters be re-formatted to increase the contrast from the black (and red) letters. Also, in the Figure 2 legend, hyphens should be added to read 'leave-one-out analysis'. In the Figure 3 legend, a hyphen should be removed to read 'leave-one-out analysis'.

We have reformatted **Figure 2A** to use blue=phased variants, red=unphased variants, and gray=homozygous variants to increase the contrast. All instances of "leave-one-out" analysis in the text and figure legends now use the same hyphenation.

5. One of the major implications of this manuscript is that it may help establish whether STRs contribute to 'missing heritability' for complex disorders and traits. In the Discussion, Reference 10 is the earliest cited article purporting that tandem repeats might be contributors to 'missing heritability', however this idea had been proposed earlier (Trends Genet. 2010; 26:59-65).

Thank you for pointing out this omission. Hannan, *et al.* (Trends Genet. 2010) is now cited in both the introduction (p. 3) and discussion (p. 12).

Reviewer #2 (Remarks to the Author):

Saini et al carried out SNP-STR phasing and impute the genotypes at STR loci. The experiments show that the procedure may increase the value of GWAS data. Some specific comments are below:

1. The concordance rate has suggested overall good performance of the pipeline. However, the concordance rate can appear more optimistic than they really is, particularly when sites are lumped together in this fashion. First of all, the concordance is highly affected by heterozygosity (which is indeed addressed in Figure 2B). It may be a good practice to remind the readers what the concordance rate would be under a random model. For example, for a bi-allelic site with allele frequencies p and q . The expected concordance would be $p^2(p^2 \cdot 1 + 2pq \cdot 0.5) + 2pq(p^2 \cdot 0.5 + 2pq \cdot 1 + q^2 \cdot 0.5) + q^2(q^2 \cdot 1 + 2pq \cdot 0.5)$. The authors used a cutoff of 5% allele frequency (heterozygosity of 0.095), which still contain lots of sites that will naturally give a high concordance rate. For example, when $p = 0.95$, the concordance would be 91% expected by random; when $p = 0.9$, the concordance rate would be 84%; when $p = 0.5$, the concordance would be 63%. So, it would be nice to show a distribution of heterozygosity for all STR sites in the paper. Then, for each heterozygosity level, indicate the expected concordance rate under a random model. This way, readers might get a better idea on how the overall concordance is comprised of, instead of lumped together. Specifically, in the Figures (Figure 2BC) and tables (Table 1 and 2), indicate where the 'random' concordance is, as a baseline to compare to.

We thank the reviewer for making this important point. Indeed, imputation performance metrics are highly affected by the allele frequency spectrum at a given STR. In the revised text, for each metric (concordance, length r^2 , and allelic r^2) we additionally compute its expectation under a random model. For concordance, this is computed as suggested by the reviewer. For the r^2 metrics, we obtained a null expectation by randomly imputing genotypes at each locus based on observed allele frequencies. We then compute length r^2 and allelic r^2 by comparing observed vs. random genotypes.

A comparison to expected metrics under a random model has been added to the following items (note some figure numbers have changed. For clarity here we provide both the old and new numberings when different):

- **Figure 2B** (heterozygosity vs. concordance).
- **Supplementary Figure 3** (previously **Figure 2C**, **Supplementary Figure 4**) (number of alleles vs. concordance, allele frequency vs. allelic r^2 , and heterozygosity vs. length r^2).
- **Table 1** (summary of imputation metrics). In addition to reporting overall average imputation metrics, we also include separate results for only multi-allelic markers and

include expected values for each metric based on a random model for comparison. (p. 17)

- **Table 2** (performance at pathogenic loci). Expected per-locus concordances are now included for comparison. (p. 18)

We additionally mention our improvement over a random model in the abstract. (p. 2)

2. Another factor which the authors could indicate in their analyses is the base pair distance from the STR site to the best tagging SNP site. Then correlate or stratify the distance in respect to concordance rate, as well as length r^2 and biallelic r^2 . This helps to get a picture on what sort of haplotype structure is expected in the phasing and imputation, and how that may affect the accuracy.

We now include the distribution of the distance between each STR and its best tag SNP (**Supplementary Figure 2B**). On average, the best tag SNP (highest length r^2) was 5.5kb away.

We additionally examined the relationship between this distance and each of our imputation performance metrics and found a modest negative relationship between this distance and imputation performance ($r=-0.06$; $p=0.06$, $r=-0.04$; $p=0.27$; and $r=-0.06$, $p=7.5 \times 10^{-5}$ for concordance, length r^2 , and allelic r^2 , respectively). (p. 7)

3. The section that claimed to show better GWAS utility is weak. It is not much of a surprise that the causal/imputed STR offers better power, since they are part of the simulation itself. Are there public phenotype/expression datasets with known associations that can be tested here and see if one can reproduce here?

As discussed above we now demonstrate the utility of imputed genotypes to discover known and novel associations of STR lengths with gene expression. Results are presented in **Figure 3D-G**. (pp. 9-10)

4. What happens to the famous Huntington's disease locus, that should also contain 'normal' alleles in the range assayable by the imputation method? However, it is not included in Table 2.

As discussed above we have expanded **Table 2** to contain additional pathogenic loci, including the Huntington's Disease locus. (p. 18)

It is true that "normal" range alleles for Huntington's Disease and other expansion disorder are mostly shorter than Illumina read lengths. However, because of HipSTR's limitation to alleles that can be spanned by short reads, it is heavily biased toward calling short alleles. Thus at loci where long alleles are frequent, returned genotypes are error prone and experience high rates of heterozygote dropout. As a result many of these loci fail our Hardy Weinberg filter. Huntington's Disease presents an additional challenge that the downstream flanking region contains an separate CCG repeat that must be spanned to obtain accurate genotypes using

HipSTR. Thus as mentioned above for these STRs we invoke a separate genotyper, Tredparse, specifically designed for targeted analysis of repeats longer than the read length.

Critical comments:

1. Probably the weakest point of the study is the lack of a true truth set, and thus the reliance on various levels of approximation: phasing is based on statistical approaches (Beagle), as it is the case also for STRs (HipSTR, LobSTR). Much of the experimentation uses simulation rather than actual physical data, or disease alleles. There is no post-hoc validation of some critical sites for the accuracy of imputation from genotyping data.

The revised manuscript provides additional validation of imputed results against capillary electrophoresis data that are not based on statistical analysis of NGS.

We do not have access to both capillary data and WGS for overlapping sets of samples used in this study. We focus in this study on validating of imputed genotypes with the assumption that high imputation accuracy implies that the phasing and genotype calling in our original haplotype panel are performing well. However, in separate studies we have (1) validated HipSTR genotypes against capillary electrophoresis data⁸, and achieved over 98% concordance after applying our standard quality filters; (2) validated Tredparse calls against capillary data for Huntington's Disease patients⁵ as well as on extensive sets of simulated genotypes. The original Tredparse paper also validates their calls against long read technologies.

We compared imputed STR genotypes vs. capillary electrophoresis data for 41 European samples at 206 highly polymorphic STRs originally from the Marshfield panel that were genotyped by the Payseur Lab. Notably, this process required converting PCR product sizes returned by capillary data to HipSTR notation, which as we have reported before can be complicated by several factors⁹ (see **Supplemental Chapter: Worldwide variation in short tandem repeats**), including "off-by-one" errors in the capillary data and polymorphic indels in flanking regions that are captured by PCR of STR regions but not counted by HipSTR.

We obtained overall concordance of 76.9% between imputed genotypes and genotypes obtained from capillary electrophoresis. For comparison, in a separate set of samples for the same loci overall concordance between HipSTR genotypes and capillary was 76.4%. Per-locus concordances were highly concordant with those based on HipSTR genotypes for a completely separate set of samples ($r=0.83$; $p=1.05 \times 10^{-53}$; $n=206$) (**Figure 2D**). (p. 8)

We additionally provide a comparison to 10X Genomics data, where we can use barcoded reads to directly obtain phased HipSTR genotypes (**Figure 2E**). This allowed us to compare allele, rather than genotype, calls to our imputed calls. We achieved 96% concordance between alleles, and genotype concordance scores were as expected based on concordance measured using leave-one-out analysis. (p. 8)

2. As the authors openly state, they filtered long STRs, and they had limited accuracy with heterozygosis. These are important consideration to its applicability for human disease.

As discussed above, we have added STRs with known disease relevance, including the Huntington's Disease locus, to our panel.

3. In some way, this approach may suffer of the same limitations of tools for the prediction of structural variation from GWAS data.

While we can achieve high imputation accuracy for most STRs, we agree that similar to structural variation some STRs will remain too difficult to impute into GWAS data. As the availability of NGS grows, eventually it will be possible to simply genotype these more complex variants directly. Still, for most traits direct analysis of complex variants from NGS is infeasible due to limited sample sizes, and imputation will allow analysis of repeats that would not otherwise be possible using hundreds of thousands of existing SNP chip datasets.

In summary, this is a good addition to the toolkit for existing GWAS studies. However, it may be suboptimally tested for actual accuracy. The value of this work would increase significantly if there were, at least, an effort at post-hoc assessment of the precision of the tool for medically relevant loci.

We thank the reviewer for the helpful comments. We believe the additional validation experiments, inclusion of medically relevant loci, and tests using real phenotype data demonstrate the utility of our panel for imputation and analysis of STRs relevant to human health.

Reviewer #3 (Remarks to the Author):

A reference haplotype panel for genome-wide imputation of short tandem repeats

This study uses genomic data from families to produce a set of reference haplotypes involving both SNPs and STRs. The authors run Beagle to impute STRs in a genome-wide dataset, obtaining a high rate of concordance between imputed and known genotypes. The paper is a milestone in human STR studies, which have long been interested in LD between STRs and SNPs. The study does for STRs much of what early haplotype map and imputation studies did for SNPs.

We thank the reviewer for the encouraging words and positive feedback. We have addressed each comment in detail below.

One issue that could be better explored is the basis for the variation in imputation accuracy. From SNP studies, the accuracy is expected to depend on the level of LD between the SNPs and STRs, the level of variation of the STRs, and the population of origin of the samples. The

authors could dig through the results a bit more to report on the impact of these various factors on the outcomes. The most difficult loci have moderate accuracy well below the 97% result in the abstract. The comparison to the CODIS imputation of Edge et al. 2017 is interesting in this regard. With their larger sample, the pedigree information, the greater population homogeneity, and the more favorable leave-one-out approach, the authors' imputation accuracies at the CODIS loci are not much higher than in Edge et al. The study is possibly near the limit of what can be achieved for some of the most highly polymorphic loci.

As the reviewer points out, various factors contribute to the relative difficulty of imputing STRs compared to SNPs. These largely include: (1) the rapid mutation rate, and corresponding high polymorphism rates of STRs and (2) a high rates of "recurrent" mutation causing the same STR allele to be present on multiple independent SNP haplotypes, in addition to population history factors that globally affect LD of all variant types. We have edited the introduction to further emphasize these points.

As we report in **Figure 2C**, and **Supplementary Figures 3-4**, the most important contributor to variation in imputation accuracy (and LD) is the level of variation of the STR resulting. STRs can have highly variable mutation rates and step sizes^{10,11}, manifesting in a wide range of heterozygosities across loci. STRs that are bi-allelic (and with slow mutation rates) are imputed with almost perfect accuracy. On the other hand, STRs with high heterozygosity (e.g. known pathogenic loci, CODIS) are much more difficult with concordances closer to 70% on average.

To clarify this, we have (1) edited the abstract and introduction to report that imputation accuracy varies across a range based on properties of the STR (2) included a comparison of all imputation metrics to values expected under a random model, which additionally demonstrate the variability of performance with polymorphism levels.

Furthermore, several figures provide specific examples of the breakdown in LD at highly polymorphic loci. **Figure 4A** demonstrates the complex relationship between SNP haplotype and STR alleles at an example pathogenic loci. **Supplementary Figure 6** shows an example of an incorrectly imputed allele at the CODIS locus D7S820 in NA12878, where we imputed 9xTATC rather than the correct call of 10xTATC. For the incorrect allele, nearly all matching SNP haplotypes had 9xTATC repeats whereas only a minority had 10xTATC, likely resulting from a recent mutation from 9 to 10 repeats.

A second comment is that the study could reach farther into the literature to make its significance clear as more than a technical advance to GWAS with STRs. Some studies along the way toward this imputation work include the early study of LD between an STR and a SNP in Tishkoff et al. 1996 Science and the SNPSTRs of Mountain et al. 2002 Genome Res. A key reference is Payseur et al. 2008 AJHG, which certainly merits some discussion as background.

We apologize for originally omitting discussion of important work leading toward STR imputation. We have now referenced previous studies of STR-SNP LD, including those mentioned by the reviewer, in the introduction. (p. 4)

Although the test the authors use for power to detect association with the repeat length of an STR is nice, they could magnify the impact by choosing a simulated trait that is connected to an STR in a more indirect way. This would evaluate the benefit of STR imputation for a more typical trait.

As detailed above, to demonstrate the power of imputation in a more realistic scenario, we now apply STR imputation to identify associations between STR lengths and gene expression using real data. Results are presented in **Figure 3D-G**. (pp. 9-10)

Abstract

Capture on average 20% more variation... compared to individual common SNPs – the exact computation and meaning of the statement are not clear here (see also p. 4).

This sentence has been removed from the abstract and clarified where mentioned elsewhere. The increase refers to the average difference in length r^2 between imputed STR genotypes vs. true STR genotypes compared to the best tag SNP vs. true STR genotypes. (p. 2)

Highlighting a limitation – the limitation is that STRs are not typically studied.

This sentence has been removed in the revised abstract. (p. 2)

Introduction

One of the largest sources of variation – clarify the sense in which this is meant.

This now reads: “STRs exhibit high rates of heterozygosity and likely contribute more *de novo* mutations per generation than all other known sources of genetic variation” (p. 3)

Genotyping errors introduced during PCR amplification – this topic goes back to early STR studies (see e.g. Lai et al. 2003 J Comp Biol and Lai & Sun 2003 J Theor Biol).

The introduction now references Lai *et al.* 2003 and Lai & Sun 2003. (p. 4)

Results

Mostly European – give the population composition with counts here.

Population composition is now given in the text and in the legend of **Supplementary Figure 1**. Based on comparison to 1000 Genomes samples, the cohort consists of 1,585 Europeans (82.7%), 39 East Asians (2.0%), 172 South Asians (9.0%), 69 Africans (3.6%), and 51 individuals (2.7%) with likely mixed ancestry. (p. 5)

Profile autosomal STRs – clarify what HipSTR does (in particular, explain the meaning of “profile”).

We have now expanded on the HipSTR method. HipSTR takes aligned reads and a reference set of STRs as input and outputs maximum likelihood diploid genotypes for each STR in the genome. While HipSTR infers the entire sequence of each STR allele, we focus here on differences in repeat copy number rather than sequence variation within the repeat itself. (p. 5)

“loci” on p. 5 – better to use “STR” or “SNP” to avoid ambiguity.

All instances of “loci” that refer to STRs specifically have been replaced in the text with “STRs”.

Heterozygosities – how are these computed? Are these observed heterozygosities or expected heterozygosities? This is ambiguous through the paper.

In all cases we compute “heterozygosity” based on observed STR genotypes. For an STR with alleles $\{1\dots n\}$, let p_i be the frequency of the i th allele computed from observed genotypes.

Then heterozygosity is defined as $H = 1 - \sum_{i=1}^n p_i^2$. Thus an STR with no variation has $H=0$,

whereas a highly polymorphic STR will have H closer to 1. Note in this case we only consider length, rather than sequence differences, when defining alleles as described below. This definition is now given in the **Online Methods**. (p. 20)

Which is not supported – this is a little awkward, I think you mean that Beagle is the only program that accommodates all the features in the sentence.

This sentence has been edited to “Based on our LD analysis, we used a window size of ± 50 kb to phase each STR separately using Beagle, which was recently demonstrated to perform well in phasing multi-allelic STRs¹³ and can incorporate pedigree information.” (pp. 6-7)

Consistent with observations from SNP imputation – it is not clear this is entirely consistent. East Asians have slightly higher LD compared to Europeans. So with all other factors being equal (e.g. size of reference panel), East Asians will be expected to have higher concordance.

This has been clarified and now reads: “Concordance was noticeably weaker in African and East Asian samples, likely due to different population background compared to the SSC samples and lower LD in African populations”. (p. 8)

Only long alleles have pathogenic effects – add reference.

We have shortened the section on power simulations in order to demonstrate imputation using a real phenotype (gene expression). The sentence referred to has been removed in the revised text.

Resolution of SNP-STR haplotypes – this type of joint SNP-STR analysis was common in earlier SNP-STR studies (e.g. Tishkoff et al. 1996 Science, Mountain et al. 2002 Genome Res, Schroeder et al. 2009 MBE).

These are now referenced in the text. (p. 11)

Discussion

In this first paragraph it would be good to give some numbers, e.g. the fraction of loci where specific concordance thresholds are achieved.

The first paragraph of the discussion now includes more detailed results about imputation performance and discusses the fact that performance varied widely based on properties of each STR. (p. 11)

Methods

Open with a description of exactly what the data are – how many samples from each of the populations, how many SNPs and STRs were detected, etc.

The **Online Methods** now begins with the section **SSC Dataset** which gives an overall description of the dataset. (p. 19)

Collapse alleles of identical length into a single allele – describe how often this heterogeneity is collapsed.

On average each length-based allele corresponded to 1.8 sequence-based alleles. This is now reported in the **Online Methods** section “Computing STR heterozygosity.” (p. 20)

Figures

Some of these plots could be presented using a more informative heat map e.g. Figs 1D, 2D, 2E, S5. Fig 7B, 8B, and 8C also could be presented without so many overlapping points.

Figure 1D, 2C (previously 2D), and **Supplementary Figure 5** (previously 2E and Supplementary Figure 5) are now presented as heatmaps.

Scatter plots in **Supplementary Figures 8** and **9** (previously Supplementary Figures 7 and 8) have been changed to bivariate kernel density plots to better depict the distribution of data points.

Fig S1. In the caption, describe the number of individuals in each group.

The legend now contains the population composition, which is listed in the comment above.

Fig S2. This plot is likely to be partially explained by the fact that r^2 is bounded above by a function of the allele frequencies at a pair of loci. The high polymorphism at the STRs is likely to lead to allele frequencies that differ between the STR and the SNP, so that the bound on r^2 is lower.

Indeed, as discussed above a major factor contributing to lower STR-SNP LD is the high polymorphism rate of STRs. Based on the suggestion from Reviewer 2, we now discuss this relationship in more detail in the text and compare all results to those expected using randomly imputed genotypes for markers of similar polymorphism levels. (e.g. **Table 1**, **Figure 2B**).

Fig S3. Make this caption more self contained, describing the method and the meaning of “concordance.”

This figure compares concordance based on the leave-one-out analysis to predicted mutation rates at each STR. The figure legend now gives a more detailed description of what “concordance” refers to and elaborates on how mutation rates were estimated.

Fig S6. Explain how the best tag SNP is found. What is the meaning of “alternate”?

Alternate allele was used to refer to any allele that is not the “major” (most frequent) allele. To avoid confusion. we have changed this figure to show the total number of STR alleles, rather than the number of “alternate alleles”. We have additionally clarified that the “best tag SNP” refers to the SNP within 50kb of the STR with the highest length r^2 .

References cited

1. Gymrek, M. *et al.* Abundant contribution of short tandem repeats to gene expression variation in humans. *Nat. Genet.* **48**, 22–29 (2016).
2. Quilez, J. *et al.* Polymorphic tandem repeats within gene promoters act as modifiers of gene expression and DNA methylation in humans. *Nucleic Acids Res.* **44**, 3750–3762 (2016).
3. Dolzhenko, E. *et al.* Detection of long repeat expansions from PCR-free whole-genome sequence data. *Genome Res.* **27**, 1895–1903 (2017).
4. Tang, H. *et al.* Profiling of Short-Tandem-Repeat Disease Alleles in 12,632 Human Whole Genomes. *Am. J. Hum. Genet.* **101**, 700–715 (2017).
5. Mousavi, N., Shleizer-Burko, S. & Gymrek, M. Profiling the genome-wide landscape of tandem repeat expansions. *bioRxiv* 361162 (2018). doi:10.1101/361162
6. Borel, C. *et al.* Tandem repeat sequence variation as causative cis-eQTLs for protein-coding gene expression variation: the case of CSTB. *Hum. Mutat.* **33**, 1302–1309 (2012).
7. Lalioti, M. D. *et al.* Dodecamer repeat expansion in cystatin B gene in progressive myoclonus epilepsy. *Nature* **386**, 847–851 (1997).
8. Willems, T. *et al.* Genome-wide profiling of heritable and de novo STR variations. *Nat. Methods* (2017). doi:10.1038/nmeth.4267
9. Mallick, S. *et al.* The Simons Genome Diversity Project: 300 genomes from 142 diverse populations. *Nature* (2016). doi:10.1038/nature18964
10. Payseur, B. A., Place, M. & Weber, J. L. Linkage disequilibrium between STRPs and SNPs across the human genome. *Am. J. Hum. Genet.* **82**, 1039–1050 (2008).
11. Gymrek, M., Willems, T., Reich, D. & Erlich, Y. Interpreting short tandem repeat variations

- in humans using mutational constraint. *Nat. Genet.* **49**, 1495–1501 (2017).
12. Browning, S. R. & Browning, B. L. Rapid and accurate haplotype phasing and missing-data inference for whole-genome association studies by use of localized haplotype clustering. *Am. J. Hum. Genet.* **81**, 1084–1097 (2007).
 13. Edge, M. D., Algee-Hewitt, B. F. B., Pemberton, T. J., Li, J. Z. & Rosenberg, N. A. Linkage disequilibrium matches forensic genetic records to disjoint genomic marker sets. *Proc. Natl. Acad. Sci. U. S. A.* **114**, 5671–5676 (2017).

Reviewer #1 (Remarks to the Author):

The authors have extensively revised the manuscript, which is much improved. They have adequately addressed my comments and suggestions.

Reviewer #2 (Remarks to the Author):

The revised manuscript has improved the validation of the STR tool.

There are two issues that could benefit from more clarity:

1. Use of a second tool (Tredparse) to improve prediction of longer repeats (excluded in HipSTR). The text here is ambiguous, as it seems to indicate that the proposed tool would require a second predictor – In particular, Table 1 is confusing as it lists 9 of 12 alleles as predicted by HipSTR.
2. Ambiguity in field of application. The abstract indicates that HipSTR contribution could be in “large-scale STR association studies using existing SNP datasets” (GWAS). However, it is unclear how HipSTR that uses reads/bam files as input, be readily converted to imputation on GWAS genotypes.

Minor:

Some notation is unclear: use of r or r^2 : “ $r=-0.06$; $p=0.06$, $r=-0.04$; $p=0.27$; and $r=-0.06$, $p=7.5 \times 10^{-5}$ for concordance, length r^2 , and allelic r^2 , respectively”.

Reviewer #3 (Remarks to the Author):

p.3 “likely contribute more de novo mutations per generation than all other known sources of variation.” This claim needs a reference or a calculation taking into account mutation rate and number of markers in different categories.

p.4 “complicated and nonlinear” – It is not clear what is meant by nonlinear, as the sentence does not describe a quantitative comparison.

p.5 “deepest catalog” – make this more precise by describing which aspects are “deepest.”

p.7 “8.8% higher” – clarify the sense of this computation. For example, is it meant for 90% compared to 80% to be an improvement of 12.5% (90/80) or 10% (90-80)?

p.7-8 Give some guidance about the expectations for the relative performance in these three different baseline data sets, and the distinctive aspects of the imputation that each offers compared to the others. Hiding some of the data and imputing the missing genotypes would be a commonly used approach, but here the approach is to use external data; explain the rationale for the particular validation computations chosen.

p.14 Part E represents allele frequency distributions rather than allele frequencies.

p.14 In part B, indicate that this is expected heterozygosity not observed heterozygosity. The revision clarifies that heterozygosity computations are expected heterozygosity, but this information could be added to the main text and figure captions.

Table 1. Split the random imputations into separate columns.

p.22 The model of random imputation is a reasonable one. However, it is not obvious that it would produce higher concordance than a model in which the imputation algorithm always imputes the

most frequent diploid genotype. A naïve concordance by this alternative approach might be greater than the values considered.

Figures. This paper has a large number of panels representing many distinct types of graphic. I would suggest that the authors take a close look at all axis labels to find as many as possible where the labels can be made more informative. For example in Fig 1A, the x-axis could be "Number of loci genotyped," and the Fig 1B x-axis could be "Number of samples with nonmissing data." In Fig 2, some of the panels are plotting concordances between concordances, and it would be better to more completely specify the axis labels, avoiding the LOO abbreviation. Fig 3 is oriented with A, B, C vertical rather than horizontal in Figs 1 and 2.

A reference haplotype panel for genome-wide imputation of short tandem repeats

Reply to reviewers

Overview

We thank the reviewers for their detailed review of the manuscript and insightful comments. We have revised our manuscript to clarify the major applications of our panel and have improved the figures as suggested to make them more easily interpretable. We have addressed each individual comment below. All changes to the main text are indicated in **red**, both here and in the revised main text.

Reviewer #1 (Remarks to the Author):

The authors have extensively revised the manuscript, which is much improved. They have adequately addressed my comments and suggestions.

We thank the reviewer for the positive feedback.

Reviewer #2 (Remarks to the Author):

The revised manuscript has improved the validation of the STR tool.

We thank the reviewer for the recognition of the improved validation of our STR reference haplotype panel.

There are two issues that could benefit from more clarity:

Use of a second tool (Tredparse) to improve prediction of longer repeats (excluded in HipSTR). The text here is ambiguous, as it seems to indicate that the proposed tool would require a second predictor – In particular, Table 1 is confusing as it list 9 of 12 alleles as predicted by HipSTR.

We have revised the text to clarify that Tredparse is used to supplement our STR catalog originally based only on HipSTR calls, which had excluded many known pathogenic STRs. Tredparse/HipSTR are used as two separate genotypers on equal footing, rather than using one to predict the other.

A potential point of confusion as raised by the reviewer is the fact that some STRs end up being called by both HipSTR and Tredparse. In all such cases the genotypes were highly concordant across tools (**Supplementary Table 1**), and we used only Tredparse genotypes for downstream analyses. We now explicitly state how this was handled when we initially describe the catalog. Subsequently we no longer treat the HipSTR and Tredparse calls separately. Specifically, we have added the following edits to clarify the use of Tredparse vs. HipSTR;

- “To supplement our panel, we applied a second STR genotyper, Tredparse, to genotype a targeted set of known pathogenic STRs in our cohort”. (p.5)
- “For seven STRs called by both Tredparse and HipSTR, Tredparse genotypes were used for downstream analysis.” (p.5)
- We have removed the partitioning of **Table 2** (which we believe the reviewer is referring to rather than **Table 1**) and **Supplementary Fig. 10** so that HipSTR and Tredparse genotypes are not treated separately.

Ambiguity in field of application. The abstract indicates that HipSTR contribution could be in “large-scale STR association studies using existing SNP datasets” (GWAS). However, it is unclear how HipSTR that uses reads/bam files as input, be readily converted to imputation on GWAS genotypes.

We apologize that the abstract was not clear on the target field of application. Indeed, the major goal of this study is to provide a resource for imputing STRs into GWAS genotypes when no NGS (reads/BAM files) are available. To clarify this point, we have edited the abstract to contain the following:

“We leverage next-generation sequencing (NGS) from 479 families to create a SNP+STR reference haplotype panel. Our panel can be used to impute STR genotypes into SNP array data when NGS is not available for directly genotyping STRs.”

We have additionally reworded several sentences in the abstract to emphasize that our panel allows imputing STRs when no sequencing data is available.

Minor:

Some notation is unclear: use of r or r^2 : “ $r=-0.06$; $p=0.06$, $r=-0.04$; $p=0.27$; and $r=-0.06$, $p=7.5 \times 10^{-5}$ for concordance, length r^2 , and allelic r^2 , respectively”.

We apologize for the unclear notation. In this case, “ r ” refers to Pearson correlation between distance to the best tag SNP and each per-STR concordance metric. We have clarified the text to use “Pearson r ” to refer to Pearson correlation, whereas “length r^2 ” and “allelic r^2 ” refer to imputation metrics as defined in the **Online Methods** (see “Imputation performance metrics”).

Reviewer #3 (Remarks to the Author):

We thank the reviewer for many helpful comments for improving our manuscript. We have noted the respective changes below.

p.3 “likely contribute more de novo mutations per generation than all other known sources of variation.” This claim needs a reference or a calculation taking into account mutation rate and number of markers in different categories.

To avoid confusion, we have revised this sentence to read “likely contribute at least as many *de novo* mutations per generation as SNPs”. We have included additional references on which we base this claim. The study by Sun et al, “A direct characterization of human mutation based on microsatellites”, states: “the mutation rate of microsatellites is around 10^{-4} to 10^{-3} per locus per generation, far higher than the nucleotide substitution rate of 10^{-8} ”. Additionally, in the study by Williams et. al, “Population-Scale Sequencing Data Enable Precise Estimates of Y-STR Mutation Rates”, we estimated mutation rates for STRs across the genome which indicated at least 75 STR mutations genome-wide per generation; whereas *de novo* SNV mutations are reported to range from 44-82 (Rocio Acuna-Hidalgo, et al. (2016)). (p.3)

p.4 “complicated and nonlinear” – It is not clear what is meant by nonlinear, as the sentence does not describe a quantitative comparison.

We have revised this sentence to more precisely explain this point. The sentence now reads: “As a result, the relationship between STR repeat number and SNP haplotype can be complex: the same STR allele may be present on multiple SNP haplotypes. On the other hand, a single SNP haplotype may harbor multiple distinct STR alleles.” (p.4)

p.5 “deepest catalog” – make this more precise by describing which aspects are “deepest.”

To avoid confusion, we have revised this sentence to “We first generated a genome-wide catalog of STR variation in a cohort of families included in the Simons Simplex Collection (SSC)”. (p.5)

“Deepest” had referred to the combination of sample size and sequencing depth. Previous catalogs were generated from low-coverage 1000 Genomes data (Willems *et al.* 2014) or from 300 high coverage diverse genomes (Mallick, *et al.* 2016). This is the first STR catalog consisting of nearly 2,000 high coverage datasets.

p.7 “8.8% higher” – clarify the sense of this computation. For example, is it meant for 90% compared to 80% to be an improvement of 12.5% (90/80) or 10% (90-80)?

We thank the reviewer for pointing out that this result can be interpreted in multiple ways. It refers to the average difference in concordance. We have added the following note in parentheses: “average concordance across STRs 69.1% vs. 60.3% using our panel vs. in Edge, *et al* “ (p.7)

p.7-8 Give some guidance about the expectations for the relative performance in these three different baseline data sets, and the distinctive aspects of the imputation that each offers compared to the others. Hiding some of the data and imputing the missing genotypes would be a commonly used approach, but here the approach is to use external data; explain the rationale for the particular validation computations chosen.

We agree that the text could benefit from clarifying the motivation for each of these validation analyses.

Indeed, the most straightforward approach is to hide some data and impute missing genotypes. We start off with the “leave-one-out” analysis which does exactly this. We then move into more extensive validation on external datasets to show that imputation using our panel is robust across different cohorts and genotyping technologies. For example, one might imagine that using SNPs obtained from WGS vs. SNP arrays, or using different genotyping pipelines, might affect imputation results. Because our ultimate goal is to impute into external GWAS data that wasn’t used to build the panel in the first place, we believe it is critical to show this is possible.

We have now added more detailed explanations motivating each of the three external datasets used for 1000 Genomes samples. The paragraph introducing these datasets has been revised as follows (new text shown in red):

“We next evaluated our ability to impute STR genotypes into external datasets. For this, we focused on samples from the 1000 Genomes Project³⁸ with high quality SNP genotypes obtained from low coverage whole genome sequencing (WGS) (n=2,504) or genotyping arrays (n=2,486 for Affy 6.0, and n=2,318 for Omni 2.5). We validated imputed genotypes for subsets of 1000 Genomes samples using data obtained from three pipelines: (1) Illumina WGS+HipSTR, (2) capillary electrophoresis, and (3) 10X Genomics+HipSTR, in each case using the orthogonal data as the “truth” set. Each of these datasets evaluates a different aspect of our imputation pipeline. The first tests whether a pipeline identical to that used to create our reference panel can achieve similar performance on datasets collected by different groups using different protocols. Additionally, since it consists of both Europeans and non-Europeans, it allows us to evaluate imputation across a variety of population groups. The second tests whether our results are robust across STR genotyping technologies and allows us to compare imputed STRs based on statistically inferred HipSTR genotypes to those obtained experimentally using capillary electrophoresis. The third returns phased data, allowing us to directly compare inferred haplotypes and phase information.” (pp. 7-8)

p.14 Part E represents allele frequency distributions rather than allele frequencies.

We have updated the **Figure 1** part E legend and the reference to the figure in the text to indicate “allele frequency distributions at pathogenic STRs”. (p. 6)

p.14 In part B, indicate that this is expected heterozygosity not observed heterozygosity. The revision clarifies that heterozygosity computations are expected heterozygosity, but this information could be added to the main text and figure captions.

We agree with the reviewer and now refer to “expected heterozygosity” throughout the text and in the figures.

Table 1. Split the random imputations into separate columns.

We have followed the reviewer's suggestion, and **Table 1** results are now broken into separate columns.

p.22 The model of random imputation is a reasonable one. However, it is not obvious that it would produce higher concordance than a model in which the imputation algorithm always imputes the most frequent diploid genotype. A naïve concordance by this alternative approach might be greater than the values considered.

Indeed, the reviewer is correct that a naive imputation algorithm that simply guesses the most common genotype achieves higher concordance compared to imputation under a random model. Notably, imputation using the SNP-STR haplotype panel still achieves far better concordance than either null model.

We have updated the abstract, main text, **Figure 2B**, and **Tables 1-2** to include the concordance achieved by imputing the most frequent diploid genotype for comparison.

Figures. This paper has a large number of panels representing many distinct types of graphic. I would suggest that the authors take a close look at all axis labels to find as many as possible where the labels can be made more informative. For example in Fig 1A, the x-axis could be "Number of loci genotyped," and the Fig 1B x-axis could be "Number of samples with nonmissing data." In Fig 2, some of the panels are plotting concordances between concordances, and it would be better to more completely specify the axis labels, avoiding the LOO abbreviation. Fig 3 is oriented with A, B, C vertical rather than horizontal in Figs 1 and 2.

We agree with the reviewer's comment and have accordingly updated the following points in the figures:

- Figure 1A: X-axis now reads "Number of STRs genotyped (millions)"
- Figure 1B: X-axis now reads "Number of samples genotyped" and y-axis now reads "Number of STRs (\log_{10})"
- Figure 1C: X-axis now reads "HipSTR Q threshold"
- Figure 1D: As suggested above, axis labels refer to "expected heterozygosity"
- Figure 2B, C, and E specify that they are from the "leave-one-out" analysis.
- Figure 2D specifies concordances are from the 1000 Genomes samples.
- Figure 3 has been rearranged such that A, B, and C are oriented horizontally. To allow "F" and "G" to be aligned, "D" and "E" are still vertically oriented.

Reviewer #2 (Remarks to the Author):

I am satisfied with the revisions - in particular, the abstract now presents a clear description of the content of the manuscript.

Reviewer #3 (Remarks to the Author):

Authors have responded nicely to the remaining concerns. It is interesting but not unexpected that the "naive" method performs better than the "random" method, though of course the main point is that the more sophisticated method performs much better than both.